# A non-canonical BRD9-containing BAF chromatin remodeling complex regulates naive pluripotency in mouse embryonic stem cells

Jovylyn Gatchalian[1], Shivani Malik[1], Josephine Ho[1], Dong-Sung Lee[2], Timothy W.R. Kelso[1], Maxim N. Shokhirev[3], Jesse R. Dixon[2] & Diana C. Hargreaves[1]

The role of individual subunits in the targeting and function of the mammalian BRG1-associated factors (BAF) complex in embryonic stem cell (ESC) pluripotency maintenance has not yet been elucidated. Here we find that the Bromodomain containing protein 9 (BRD9) and Glioma tumor suppressor candidate region gene 1 (GLTSCR1) or its paralog GLTSCR1-like (GLTSCR1L) define a smaller, non-canonical BAF complex (GBAF complex) in mouse ESCs that is distinct from the canonical ESC BAF complex (esBAF). GBAF and esBAF complexes are targeted to different genomic features, with GBAF co-localizing with key regulators of naive pluripotency, which is consistent with its specific function in maintaining naive pluripotency gene expression. BRD9 interacts with BRD4 in a bromodomain-dependent fashion, which leads to the recruitment of GBAF complexes to chromatin, explaining the functional similarity between these epigenetic regulators. Together, our results highlight the biological importance of BAF complex heterogeneity in maintaining the transcriptional network of pluripotency.

[1] Molecular and Cellular Biology Laboratory, Salk Institute for Biological Studies, 10010 N. Torrey Pines Rd, La Jolla, CA 92037, USA. [2] Peptide Biology Laboratory, Salk Institute for Biological Studies, 10010 N. Torrey Pines Rd, La Jolla, CA 92037, USA. [3] Razavi Newman Integrative Genomics and Bioinformatics Core, Salk Institute for Biological Studies, 10010 N. Torrey Pines Rd, La Jolla, CA 92037, USA. Correspondence and requests for materials should be addressed to D.C.H. (email: dhargreaves@salk.edu)

Embryonic stem cells (ESCs) have the remarkable ability to self-renew and give rise to any of the over 200 different mature cell types, the property of pluripotency. To maintain ESC identity, the genome is precisely controlled so that only stem cell-specific transcription programs are turned on while lineage-specific programs are silenced[1,2]. This control is in part achieved by ATP-dependent chromatin remodeling complexes, which regulate chromatin structure[3,4]. In particular, several subunits of the mammalian BRG1-associated factors (BAF) chromatin remodeling complex are required for formation of the inner cell mass (ICM) of the embryo and for maintenance of ESCs in vitro[5–8].

In mouse ESCs, a specialized BAF complex exists in the form of esBAF, a ~2 MDa complex composed of 9–11 subunits including BRG1, BAF155, BAF47, ARID1A, BAF45A/D, BAF53A, BAF57, SS18, and BAF60A/C[9,10]. Genome sequencing studies using conditional deletion of BRG1, the ATPase component of the complex, revealed that esBAF collaborates with the master pluripotency regulators OCT4, SOX2, and NANOG in modulating the expression of ESC-specific genes while repressing genes associated with differentiation[11,12]. In addition to esBAF, the related polybromo-associated BAF (PBAF) complex is also present, which contains distinct components including BRD7, ARID2, and PBRM1[13–15]. Upon differentiation, the BAF complex becomes even more diversified, assembling the different subunits in a combinatorial manner, which imparts to the complex its cell type- and developmental stage-specific activities[16–18]. However, it is unclear how distinct BAF complex assemblies and the unique subunits therein contribute to BAF-dependent functions.

Mass spectrometric studies in mouse ESCs identified BRD9 as a novel BRG1-interacting partner[9] and was subsequently shown to be a dedicated BAF complex subunit in a leukemic cancer cell line[19]. BRD9 harbors a single bromodomain (BD), an epigenetic reader domain that recognizes acetylated lysines on histones and non-histone proteins[20,21]. However, the role of BRD9 in BAF complex targeting and activity remains uncharacterized. Here we find that in mouse ESCs, BRD9 and GLTSCR1/1L are defining members of the smaller, non-canonical GLTSCR1/1L-containing BAF complex or GBAF complex[22]. We perform IP-mass spectrometry characterization of the GBAF complex in mouse ESCs to define shared and unique subunits. GBAF is distinct from esBAF as it lacks BAF47, BAF57, and ARID1A. Chromatin IP (ChIP)-Seq analyses show that esBAF and GBAF are uniquely targeted to sites across the genome and are co-bound with distinct sets of pluripotency transcription factors (TFs). The genomic binding of the GBAF complex is consistent with its role in maintaining the naive pluripotent state, as inhibition of BRD9 results in transcriptional changes representative of a primed epiblast-like state. Conditional deletion of esBAF subunit ARID1A is not highly correlated with this transition, indicating that GBAF complexes have a functionally specific role in regulating this pathway. We demonstrate that BRD9 is targeted to chromatin via its BD, highlighting the role of this reader domain in GBAF complex targeting. Finally, we provide evidence for BD and extra-terminal protein 4 (BRD4)-mediated targeting of the GBAF complex that is BD-dependent, which accounts for their complementary roles in regulating the naive pluripotency transcriptional network. Our studies not only provide important insight into BRD9 function but also add to our understanding of how diversity in BAF complex assembly contributes to the intricate control of the ESC transcription program.

## Results

**BRD9 regulates transcription for the maintenance of ESCs.** To determine the specific role of BRD9 in the maintenance of ESC pluripotency, we made use of the BRD9-BD inhibitor, I-BRD9, which specifically inhibits binding of the BRD9-BD to acetylated residues[23]. By assaying for cell number at different time points, we found a time- and I-BRD9 concentration-dependent decrease in cell proliferation, pointing to a BD-dependent role for BRD9 in maintaining mouse ESCs in serum/leukemia inhibitory factor (LIF) conditions (Fig. 1a). Because I-BRD9 has been shown to also inhibit BRD4 in vitro[23], we tested two other BRD9-BD inhibitors (BRD9i), BI-9564 and TP472, which also yielded similar concentration-dependent growth defects in ESCs (Fig. 1b and Supplementary Figure 1a). Consistent with the specific activity of the BRD9i, short hairpin RNA (shRNA)-mediated knockdown of Brd9 in mouse ESCs with three independent shRNAs resulted in a near complete loss of BRD9 protein expression relative to that of a scrambled control (Supplementary Figures 1b, 1c), leading to significant reduction in ESC growth at 2 and 4 days post shRNA transduction (Fig. 1c).

We next addressed whether BRD9 exerts its function in pluripotency by regulating gene expression. We treated ESCs with I-BRD9 and assessed mRNA expression changes at 24 and 48 h post treatment using high-throughput sequencing (RNA-Seq). We observed dramatic changes in gene expression upon treatment with I-BRD9 (Fig. 1d; 1.5-fold change (FC), false discovery rate (FDR) < 0.05, Benjamin-Hochberg). At 24 h, I-BRD9 treatment resulted in 351 differentially expressed genes (DEGs), the vast majority of which were also present among the 929 DEGs changed following 48 h of I-BRD9 treatment (Fig. 1e). In both instances, we observed more downregulated than upregulated genes, suggesting that BRD9 generally maintains gene expression. Gene ontology (GO) analysis of I-BRD9-dependent genes revealed that BRD9 functions primarily in regulating tissue development and cellular differentiation (Fig. 1f). To confirm the on-target effects of I-BRD9, we performed RNA-Seq in ESCs where Brd9 was knocked down using a pooled collection of three shRNAs. We observed a high degree of concordance between the mRNA expression changes with I-BRD9 treatment and Brd9 knockdown. Gene set enrichment analysis (GSEA) showed that I-BRD9-downregulated genes are positively enriched among genes that decrease upon Brd9 knockdown, while I-BRD9-upregulated genes are enriched among genes that increase upon Brd9 knockdown (Fig. 1g). Conversely, the change in gene expression upon Brd9 knockdown was strongly correlated with the change upon I-BRD9 treatment (Fig. 1h). Finally, to determine whether BRD9's role in transcriptional regulation is in the context of BAF complexes, we performed GSEA to compare our I-BRD9 RNA-Seq data with a publicly available dataset of BRG1-dependent genes in mouse ESCs[12]. We saw a significant enrichment of genes downregulated by I-BRD9 in the BRG1 dataset (Fig. 1i). Consistent with this, inhibition of BRG1 activity by the small-molecule inhibitor PFI-3 caused a similar decrease in ESC cell growth (Supplementary Figure 1d). These data indicate that BRD9 and BRG1 cooperate to regulate a subset of genes in ESCs that are critical for ESC maintenance.

**BRD9 associates with a non-canonical BAF complex in ESCs.** Given the functional impact of BRD9 loss or inhibition on ESC pluripotency, we sought to further characterize BRD9's association with BAF complexes in mouse ESCs. To this end, we performed IP of endogenous BRD9 from micrococcal nuclease-treated nuclear lysates under high stringency wash conditions and analyzed the precipitated proteins by mass spectrometry (Supplementary Table 1). The results validated previous observations that BRD9 interacts with BRG1, BAF155, the BCL7 proteins, and SS18 (Fig. 2a, Supplementary Table 1)[19]. We identified other BAF

subunits, including BAF60A, BAF155, BAF53A, GLTSCR1L, and GLTSCR1, which were among the top hits in the mass spectrometry results. Surprisingly, however, we did not recover established BAF/PBAF subunits ARID1A, BAF47, BAF57, PBRM1, or BRD7, suggesting that we had identified a non-canonical BRD9-containing BAF complex.

We next isolated ESC nuclear extracts and subjected the proteins to a glycerol gradient density sedimentation assay. Our results show that BRD9 co-sediments earlier in the gradient with

BRG1, GLTSCR1, BAF155, and BAF60A, indicating that it associates with the lower molecular weight GBAF complex (Fig. 2b). BRD9 did not sediment with esBAF subunits, defined by the presence of ARID1A, BAF57, and BAF47, or with PBAF, which uniquely incorporates PBRM1 and BRD7. IP experiments with specific antibodies against BRG1, BRD9, and BAF47 verified that BRD9 is in a BAF complex distinct from esBAF, as it does not interact with ARID1A, BAF57, or BAF47 (Fig. 2c). The reciprocal IP demonstrated that BAF47 associates with known

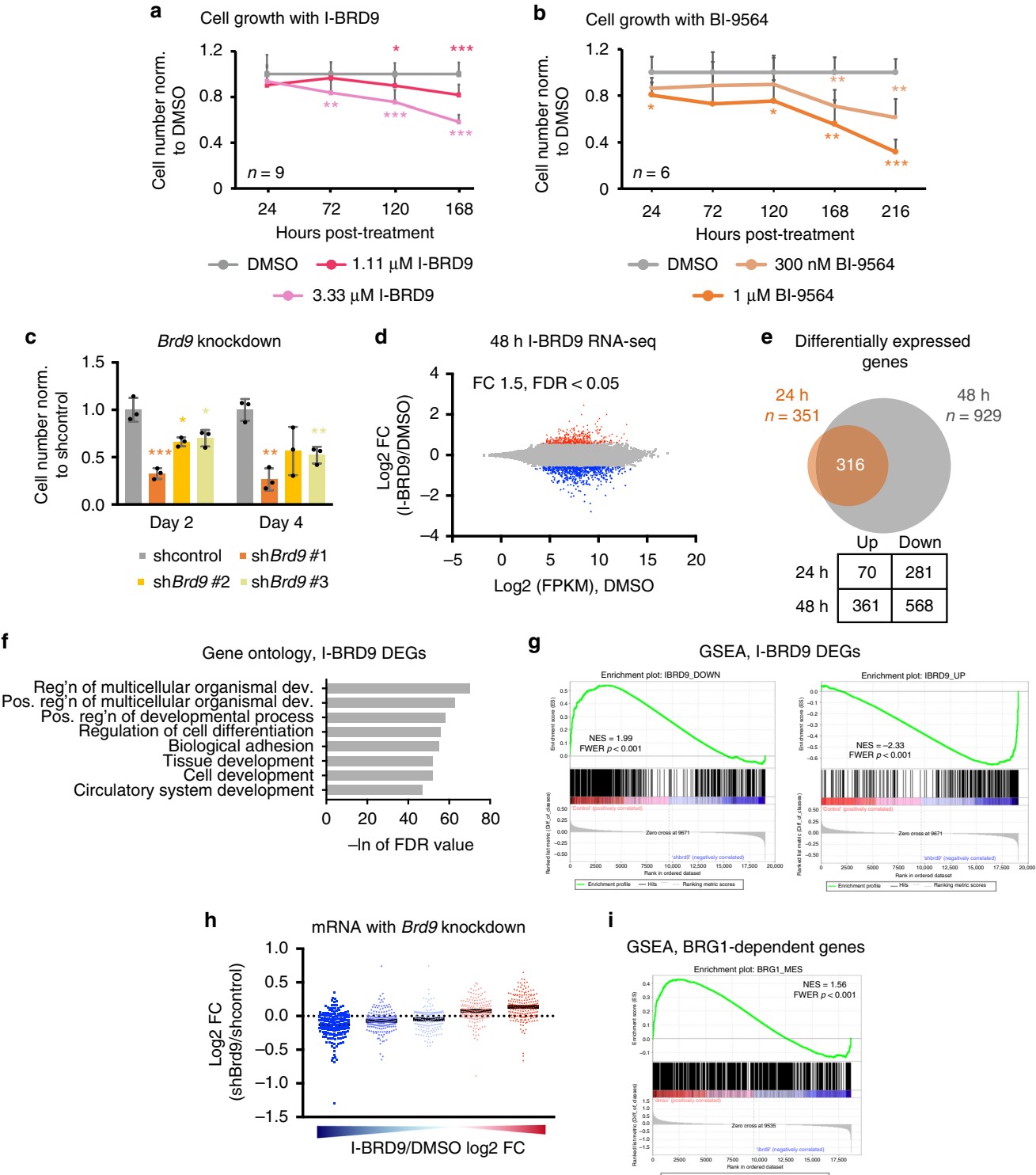

esBAF subunits, but does not interact with BRD9 (Fig. 2c). In addition, we confirmed that GLTSCR1 and GLTSCR1L each exclusively associate with BRD9, but not with BAF47 (Fig. 2c).

To demonstrate that BRD9 is a dedicated GBAF subunit, we depleted ESC nuclear lysates of either BRG1 or BAF155 using specific antibodies and monitored BRD9 levels after each round of four IP reactions. This showed that BRD9 is quickly depleted with BRG1 or BAF155, suggesting that BRD9 is exclusive to BAF complexes (Fig. 2d). Finally, we tested how stable the associations are between BRD9 and its interacting proteins using urea-based denaturation studies. We found that BRG1, BAF60A, and GLTSCR1L remain associated with BRD9 in up to 1 M urea (Fig. 2e). In summary, our biochemical studies demonstrate that BRD9 forms a stable BAF complex that is distinct from the canonical esBAF and PBAF complexes and uniquely contains GLTSCR1/1L in mouse ESCs (Fig. 2f). The presence of this non-canonical BAF complex is not unique to ESCs, as we also observed it in HCT116 cells, a human colorectal cancer line (Supplementary Figures 2a, 2b). Moreover, a recent study showed that BRD9 associates with a smaller GLTSCR1/1L-containing BAF complex termed the "GBAF complex" in a human monocytic cell line and HEK293Ts[22], which is also supported by mass spectrometry studies defining the assembly of this non-canonical complex in HEK293T cells, synovial sarcoma, and malignant rhabdoid cell lines[24]. Collectively, these results argue that the presence of GBAF is a general phenomenon as it is found in both human cancer cells and mouse ESCs, the latter of which represents a karyotype normal background. For continuity, and to distinguish it from esBAF and PBAF complexes, we will hence refer to the non-canonical BAF complex as the GBAF complex.

**GBAF and esBAF localize to distinct genomic elements.** To identify the genomic localization of the non-canonical GBAF complex in ESCs, we employed ChIP-Seq with antibodies against BRD9 and BRG1. We identified 35,932 BRD9-bound sites and 73,011 BRG1-bound sites (Fig. 3a). Eighty-one percent of sites occupied by BRD9 are also bound by BRG1 and their localization on the genome is strikingly similar, consistent with a strong biochemical interaction (Fig. 3b, c). To establish whether GBAF localizes distinctly from esBAF, we also performed ChIP-Seq against ARID1A, an esBAF subunit which is not present in GBAF. We identified 20,677 ARID1A-bound sites, which showed 72% overlap with BRG1 targets (Fig. 3a–c). Only 33% of ARID1A peaks overlapped with BRD9, suggesting that these complexes are independently targeted, but are co-bound at some sites, in agreement with several studies showing co-localization of chromatin remodeling complexes on the genome[25,26]. Co-occurrence binding analysis with publicly available histone modification data in mouse ESCs revealed a difference between BRD9 and ARID1A

targets. Specifically, we found that BRD9 is more enriched at sites that are marked with H3K4me3, an epigenetic mark more commonly found at promoters (Fig. 3d). ARID1A, on the other hand, showed stronger enrichment at sites marked with H3K4me, which is strongly associated with enhancers. Both ARID1A and BRD9 are bound at sites marked with H3K27ac, but not H3K27me3. We further examined BRD9 and ARID1A localization to different enhancer classes by defining poised (H3K4me-positive, H3K27ac-negative), active (H3K4me- and H3K27ac-positive), and super enhancers (defined by H3K4me-positive, H3K27ac-positive, and high Mediator binding[27]). Here we observed similar enrichment of ARID1A and BRD9 at active enhancers whereas at both poised and super enhancers, ARID1A showed a relatively stronger enrichment than BRD9 (Fig. 3d). BRG1 is known to bind to both promoters and distal sites[11] and further analysis of BRD9 and ARID1A binding at these different regions showed that the two have different binding proclivities, with BRD9 binding more strongly to promoters and ARID1A more strongly to distal sites (Fig. 3e). Furthermore, BRD9 and BRG1, but not ARID1A, localize to topologically associating domain (TAD) boundaries, which are also strongly enriched for H3K4me3 (Fig. 3f)[28]. These data suggest that esBAF and GBAF are preferentially targeted to different regions of the genome.

**GBAF and esBAF are co-bound with different factors.** To understand how GBAF complexes participate in the pluripotency regulatory network, we profiled GBAF and esBAF complex binding sites for enriched TF-binding motifs. We distinguished BRG1-bound sites that uniquely contain either BRD9 (GBAF) or ARID1A (esBAF) and determined the enriched motifs for both (Fig. 4a). Consistent with previous reports demonstrating the role of BRG1 in OCT4-dependent transcription[12], the motif for the master pluripotency regulators OCT4/SOX2/TCF/NANOG was the most enriched motif for esBAF complex binding, followed by motifs for the high-mobility group domain-containing Sox family members. In contrast, GBAF complex sites are enriched for the CTCF/CTCFL motif, followed by the zinc finger-containing TFs Kruppel like factor 3 (KLF3) and the Specificity proteins Sp5 and Sp1. TF binding analyses verified the results of the motif search, where we observed greater occupancy of OCT4, SOX2, and NANOG on esBAF sites while CTCF binding was stronger at GBAF sites (Fig. 4b, c). KLF4 and Sp5 binding was observed at both esBAF and GBAF binding sites, with slightly greater occupancy at GBAF complex binding sites. These data are consistent with prior literature detailing the role of esBAF complexes in regulating the master pluripotency TFs, and further highlight the specific binding pattern of GBAF complexes, suggesting cooperation between GBAF complexes and KLF4 and Sp5.

---

**Fig. 1** BRD9 is part of the ESC pluripotency transcriptional regulatory network. **a** Time course experiment assessing mouse ES cell number after treatment with DMSO or I-BRD9 at either 1.11 or 3.33 μM. Error bars represent one standard deviation from the mean of biological replicates, $n = 9$. Two-tailed $t$-test was performed to obtain the $p$ values. *$p < 0.05$, **$p < 0.01$, ***$p < 0.001$. **b** As in **a**, but for ESCs treated with DMSO or BI-9564 at either 300 nM or 1 μM; $n = 6$. **c** As in **a**, but for ESCs transduced with shRNA against a scrambled control or Brd9; $n = 3$ for each shBrd9 experiment. **d** Scatterplot of average log2 fragments per kilobase of transcript per million mapped reads (FPKM) mRNA expression level in DMSO-treated ESCs against log2 fold change (FC) in expression after 24 or 48 h of 10 μM I-BRD9 treatment in serum/LIF conditions. Red and blue indicate differential expression increased or decreased by 1.5-fold or more (Benjamin-Hochberg FDR < 0.05), respectively, from two independent biological replicates. **e** Venn diagram of differentially expressed genes (DEGs) in the 24 and 48 h I-BRD9 RNA-Seq datasets. **f** Significance of the I-BRD9 DEG enrichment in each gene ontology process. FDR values were calculated according to Benjamin-Hochberg multiple testing (GSEA). **g** GSEA enrichment plots of either the significantly downregulated (left) or upregulated (right) I-BRD9 DEGs using the shcontrol and shBrd9 RNA-Seq dataset. NES normalized enrichment score, FWER $p$ value familywise error rate calculated using single tail tests on the positive or negative side of the null distribution. **h** Box plot of the log2 FC in expression upon shBrd9 knockdown, grouped into quintiles according to the genes' log2 FC upon I-BRD9 addition. Shades of blue and red correspond to the degree of log2 FC down- or upregulation in I-BRD9/DMSO. Black bars indicate the mean per quintile. **i** As in **g**, GSEA enrichment plot of BRG1-dependent genes in ESCs using the I-BRD9 RNA-Seq dataset. Source data for **a–c** are provided as Source Data file

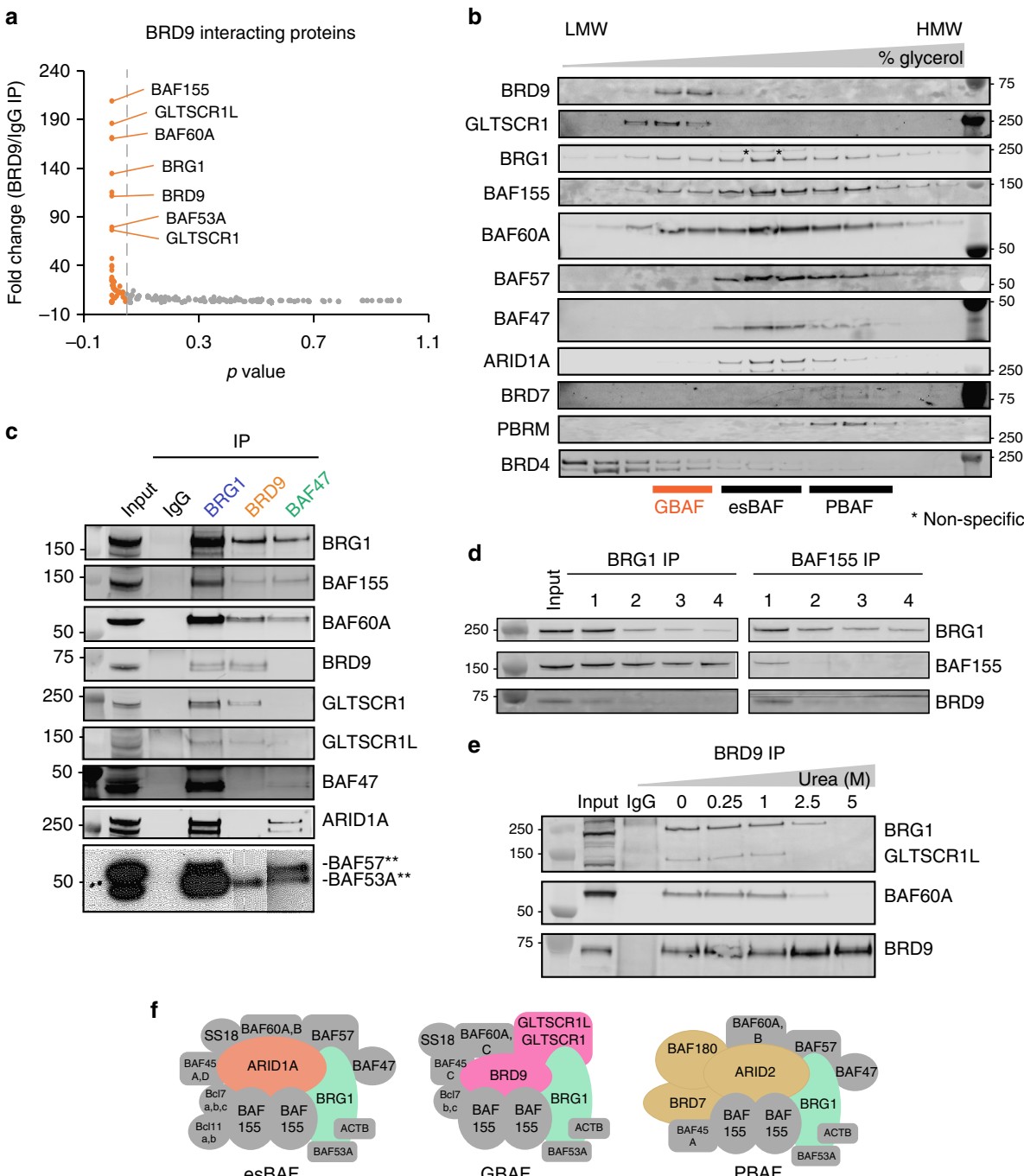

**Fig. 2** BRD9 defines a distinct, non-canonical BAF complex. **a** Immunoprecipitation (IP)-mass spectrometry using BRD9 or IgG antibody from mouse ESCs. The plot shows the spectral count fold change in BRD9 IP relative to IgG and the corresponding AC test *p* values, calculated with two technical replicates using PatternLab. In orange are proteins that satisfy FC 2, AC test *p* < 0.05. **b** Immunoblotting analysis of fractions after mouse ESC nuclear lysates were subjected to a density sedimentation assay in 10–30% glycerol gradient. LMW and HMW indicate lower and higher molecular weights, respectively. Nonspecific bands are marked with an asterisk. Molecular weights from ladder are indicated. **c** Immunoprecipitation (IP) experiments from mouse ESC nuclear lysates using antibodies against BRG1, BRD9, and BAF47. Blots developed using chemiluminescence are marked with double asterisks; each IP was taken from a different exposure. **d** Depletion IP experiment from ESCs using antibodies against BRG1 or BAF155. Each lane shows the remaining proteins after each successive IP, labeled 1 through 4. **e** IP experiment from ESCs using antibodies against IgG or BRD9, which was incubated in increasing concentration of urea. **f** Schematic of the esBAF, GBAF, and PBAF complexes in mouse ESCs. Colored in orange, pink, and mustard yellow are subunits that define each individual complex. In green is the enzymatic component, BRG1. Source data are provided as a Source Data file

**GBAF complexes regulate naive pluripotency.** Both KLF4 and Sp5 have been shown to regulate naive pluripotency in part through the transcriptional regulation of *Nanog*, whose downregulation marks the exit of ESCs from the naive state into a state that is primed for lineage specification[29–31]. Interestingly, we

observed downregulation of both *Nanog* and *Klf4* in I-BRD9-treated ESCs (Fig. 5a). To determine if this downregulation was specific to GBAF complexes, we performed RNA-Seq on *Arid1a*[f/f]:CreERT2 ESCs treated with ethanol (vehicle) or tamoxifen to induce deletion of *Arid1a*. In contrast to I-BRD9 treatment,

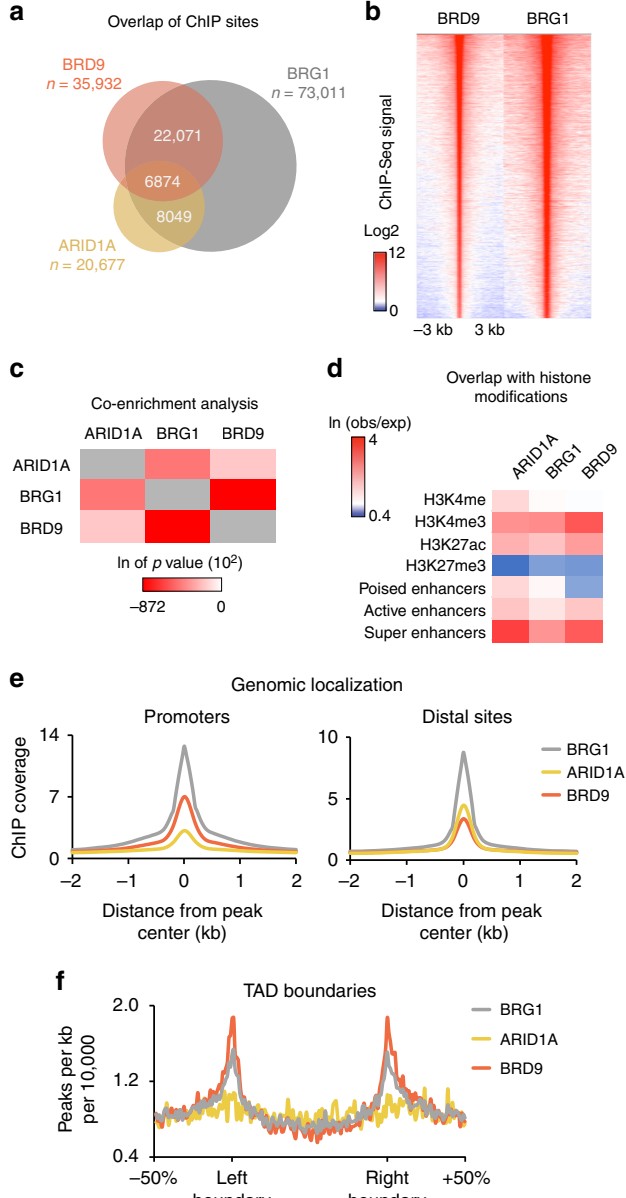

**Fig. 3** GBAF and esBAF localize to distinct genomic elements. **a** Venn diagram overlap of BRG1, BRD9, and ARID1A ChIP sites, with $n$ representing the number of observed peaks. **b** Heat map of BRD9 and BRG1 ChIP signal ± 3 kilobases (kb) centered on the BRD9 peak, ranked according to BRD9 read density. **c** Significance of binding overlap between BRD9, BRG1, and ARID1A ChIP sites. The natural log of $p$ values were calculated using hypergeometric distribution (mergePeaks.pl in HOMER). **d** Co-occurrence analysis showing the natural log of the ratio of the observed number of overlapping peaks over the expected values. This was done for BRD9, BRG1, and ARID1A ChIP sites against publicly available ChIP-Seq data for the histone modifications H3K4me, H3K4me3, H3K27me3, and H3K27ac. Poised enhancers are defined by sites that contain H3K4me1 but lack H3K27ac, whereas active enhancers are sites that are positive for both modifications. Super enhancers are H3K4me+, H3K27ac+, and Med1-high sites. **e** ChIP-Seq signal of BRG1, BRD9, and ARID1A ± 2 kb surrounding BRG1 peak center at promoters and distal sites. **f** ChIP-Seq signal of BRG1, BRD9, and ARID1A ± 50% size of a topologically associating domain (TAD) around a TAD

ARID1A loss did not affect the expression of either *Nanog* or *Klf4* (Fig. 5a). Neither BRD9 inhibition nor ARID1A deletion resulted in significant changes in the core regulators of pluripotency, *Pou5f1* or *Sox2*, consistent with what has been observed previously in BRG1-deficient ESCs[11,12]. These data suggested that GBAF complexes may specifically regulate naive pluripotency. Consistent with this, we observed that I-BRD9-treated ESCs have a flatter morphology that resembles primed or epiblast ESCs (EpiESCs)[32,33], whereas those treated with vehicle maintain the characteristic domed structure of naive ESCs (Fig. 5b). Furthermore, ESCs cultured with either I-BRD9 or BI-9564 for 6 days have reduced colony-forming activity (Fig. 5c) and yield less cells with alkaline phosphatase (AP) activity (Fig. 5d, e), which is consistent with EpiESCs being less clonogenic in culture than their naive counterparts[34]. Together, our functional data indicate that BRD9 has an important role in maintaining the naive pluripotent state.

To assess this directly, we measured the overlap between I-BRD9 DEGs and a previously published dataset comparing naive mouse ESCs cultured in serum/LIF conditions and EpiESCs generated by culturing in Activin A/FGF4[35]. We found that over half of I-BRD9 DEGs ($n = 477$) were present in this EpiESC-ESC dataset (Fig. 5f). Importantly, there was a significant correlation between genes that were upregulated or downregulated in both Activin A/FGF4 and I-BRD9 conditions (Fig. 5g, $R^2 = 0.649$, linear regression). In contrast, there was no correlation between the transcriptional changes that occur in the ESC-EpiESC transition and following deletion of *Arid1a* (Fig. 5h, $R^2 = 0.009$, linear regression). Thus, BRD9, but not ARID1A, specifically protects naive pluripotency as inhibition of BRD9 results in gene expression changes that closely resemble the primed epiblast-like state.

To determine if GBAF complexes regulate the naive pluripotency program through facilitating Sp5 and KLF4, we profiled the occupancy of these factors at genes regulated during the ESC-EpiESC transition. Consistent with the role of KLF4 and Sp5 in maintaining naive pluripotency, we found that ESC-EpiESC genes are significantly occupied by these factors, but not c-Myc (p = 0.057, hypergeometric test), in ESCs (Fig. 5i). In addition, BRD9 and BRG1 binding are highly significant at these genes, with 82% of ESC-EpiESC genes being co-bound by both. Moreover, BRD9 and BRG1 are frequently co-bound with KLF4 (88% of KLF4 targets[36]) and Sp5 (88% of Sp5 targets[37]), consistent with the enrichment of KLF4 and Sp5 motifs at GBAF complex binding sites. KLF4/Sp5-bound gene targets include key regulators of naive pluripotency, such as *Nanog*, *Nr5a2*, *Prdm14*, *Gli2*, *Tet2*, *Tdgf1*, *Fgf4*, and *Tcl1*. As expected, these genes are significantly reduced in the ESC-EpiESC transition (Fig. 5j). Moreover, these genes are also significantly downregulated by I-BRD9 treatment, indicating that GBAF complexes are required for expression of KLF4/Sp5-bound targets (Fig. 5j). Several KLF4-bound genes in the pro-differentiation FGF/MEK/ERK pathway were also upregulated in both datasets, including *Jun*, *Fosl2*, and *Atf3*, further highlighting the role of BRD9 in collaborating with KLF4 to both maintain the ESC naive state and inhibit ESC priming. These data demonstrate that BRD9 is required for the proper transcription of KLF4/Sp5-dependent targets, consistent with significant overlap of GBAF complex binding with KLF4 and Sp5 in ESCs. In contrast, conditional deletion of *Arid1a* had variable, and often minimal, effect on the expression of these genes. Thus, GBAF complexes have a functionally specific role in preserving the naive pluripotency of ESCs and preventing transition to the primed state, a function that is not controlled by canonical esBAF complexes.

**a**     Enriched motifs

ARID1AxBRG1-unique (*n* = 8049)

1. OCT4-SOX2-TCF-Nanog 26.4 % of target 1.9 % of background *p* = e −1714

2. SOX3 52.0 % of target 19.1 % of background *p* = e −944

3. SOX2 34.7 % of target 9.7 % of background *p* = e −804

4. SOX6 44.3 % of target 17.2 % of background *p* = e −690

BRD9xBRG1-unique (*n* = 22,071)

1. CTCF or CTCFL 36.8 – 39.9 % of target 1.5 – 3.2 % of background *p* = e −8696/−6948

2. KLF3 23.5 % of target 10.3 % of background *p* = e −696

3. Sp5 36.0 % of target 19.8 % of background *p* = e −689

4. Sp1 19.2 % of target 8.0 % of background *p* = e −593

**b**     Transcription factor binding

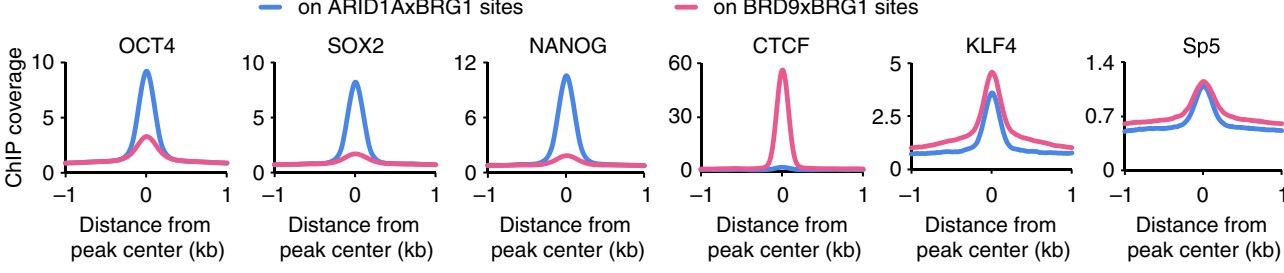

**c**     Genome browser tracks

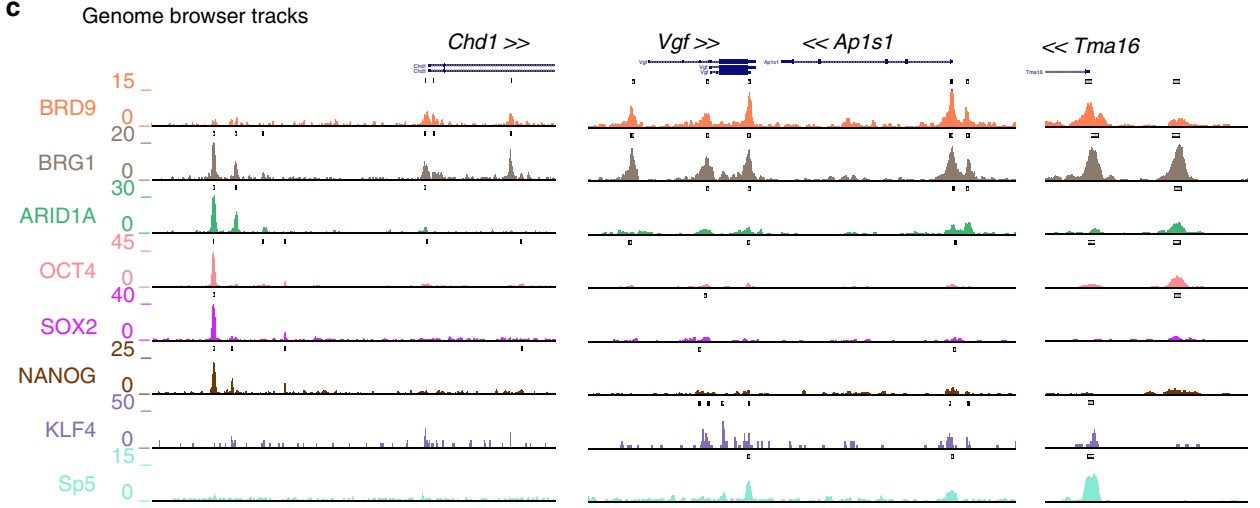

**Fig. 4** GBAF and esBAF are co-bound by different pluripotency transcription factors. **a** Significance of enriched motifs for BRG1 sites uniquely bound by either ARID1A (top) or BRD9 (bottom). *p* Values were calculated using cumulative binomial distribution (findMotifsGenome.pl in HOMER). **b** Histogram of ChIP reads of the indicated transcription factor (TF) ± 1 kb surrounding the peak center of sites that are co-bound by either ARID1A and BRG1 or BRD9 and BRG1. **c** Representative genome browser tracks showing co-binding of BRD9 and BRG1 with either KLF4 or Sp5 at promoters (middle and right, respectively) or ARID1A and BRG1 with OCT4, SOX2, and NANOG at distal sites (left)

**The BD localizes BRD9 to its genomic targets**. The BD is a well-studied reader domain that recognizes acetylated lysines on histones and non-histone proteins. BRD9's BD has been shown to bind acetylated histone peptides in vitro, with no clear preference, but its physiological target remains elusive[20]. Therefore, it is unclear whether the BD serves as a targeting module for BRD9. To address this, we performed a cellular fractionation assay from ESCs treated with I-BRD9 for 24 h to inhibit any BD-acetylated lysine interaction. We observed a global reduction of BRD9 from the chromatin fraction that was dependent on I-BRD9 concentration (Fig. 6a, b). IP experiments against BRD9 with or without I-BRD9 show that GBAF remains intact (Supplementary Figures 3a, 3b). We next performed ChIP-Seq against BRD9 24 h after treating ESCs with I-BRD9. Consistent with the fractionation assay results, BRD9 binding was markedly diminished upon inhibitor treatment (Fig. 6c). In fact, at over 12,000 sites, BRD9 occupancy was significantly reduced by at least 1.5-fold (Fig. 6d, Poisson *p* value < 0.0001), while <200 sites exhibited increased occupancy. The displacement from chromatin is rapid, as we observed loss of BRD9 after 6 h of I-BRD9 treatment at 2226 sites, 61% of which remained down at 24 h (Fig. 6e). Furthermore, the average BRD9 read density at all of its targets progressively

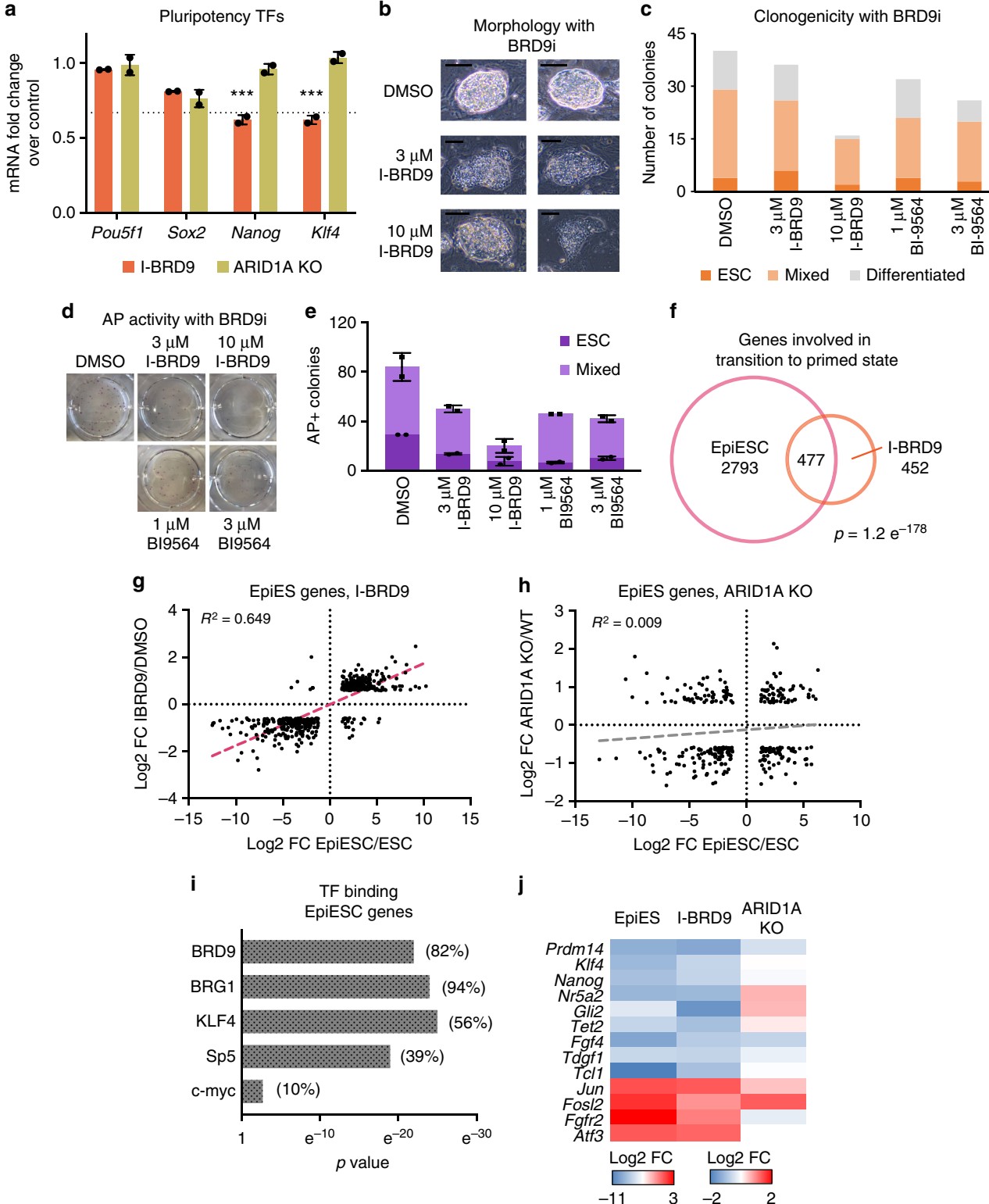

decreased over time with I-BRD9 treatment (Fig. 6f, g), in support of the BD functioning as a targeting module for BRD9.

We next assessed the effect of BRD9 displacement from chromatin on gene expression. We found that gene expression is affected at genes that exhibit loss in BRD9 occupancy upon I-BRD9 treatment (Fig. 6h). Conversely, BRD9 ChIP read density is decreased with I-BRD9 treatment at essentially all sites annotated to genes that are significantly regulated by I-BRD9 (Fig. 6i). We

also performed ChIP-Seq against BRG1 in ESCs after 24 h of I-BRD9 treatment to determine whether BRG1 binding is affected. While there was no global loss of BRG1 from chromatin with I-BRD9 by western blot (Fig. 6a and Supplementary Figure 3c), BRG1 binding at sites annotated to the I-BRD9 DEGs trended downward (Fig. 6j, k). Based on the fact that only 17% of total BRG1 is contained in GBAF complexes (Fig. 2b), we cannot rule out esBAF and PBAF binding at these sites or other

**Fig. 5** GBAF complexes regulate naive pluripotency. **a** Bar graph of mean mRNA expression FC for the pluripotency TFs upon either I-BRD9 treatment of v6.5 ESCs or tamoxifen treatment of $Arid1a^{f/f}$CreERT2 ESCs (ARID1A KO) over DMSO or ethanol, respectively, from two biological replicates. 1.5-FC decrease is denoted with a dotted line. Error bar = standard deviation. Benjamin-Hochberg FDR < 0.05 is denoted with triple asterisks. **b** Brightfield images of ESCs treated with either DMSO or I-BRD9 at 3 or 10 µM for 6 days. Scale bar = 150 µm. **c** Quantification of ESC colonies cultured with the indicated treatment for 6 days. This is representative of two independent experiments. **d** Representative images of wells containing ESCs treated with the indicated vehicle or BRD9i for 6 days then assayed for alkaline phosphatase (AP) activity. **e** Quantification of the colonies in **d**, showing the mean and standard deviation of two biological replicates. Error bars = standard deviation. Source data are provided as Source Data file. **f** Venn diagram overlap of DEGs with I-BRD9 treatment for 48 h and DEGs with FGF/Activin A addition over serum/LIF (EpiESC/ESC). $p$ value was calculated using hypergeometric test of overlap, with population size being the total number of genes tested ($N = 24{,}538$). **g** Scatterplot of the mRNA log2 FCs in I-BRD9/DMSO and EpiESC/ESC for the 477 common DEGs in **f**. Linear regression analysis was performed to calculate the $R^2$. Best fit is represented as a pink dashed line. **h** As in **g**, but of the mRNA log2 FC of DEGs that are common between $Arid1a^{f/f}$CreERT2 ESCs treated with tamoxifen/ethanol and EpiESC/ESC ($n = 302$). Best fit is represented as a gray dashed line. **i** Significance of TF binding on the 477 common DEGs in **f**. $p$ values were calculated using hypergeometric test. In parentheses are the percentages of EpiESC genes that are bound by the corresponding TFs. **j** Heat map of mRNA log2 FCs for the indicated genes in EpiESC/ESC, I-BRD9/DMSO, and $Arid1a^{f/f}$CreERT2 ESCs tamoxifen/ethanolSource data

compensatory mechanisms. Altogether, our data demonstrate that the BD is essential for targeting BRD9, and is required for BRD9's role in gene expression.

**BRD4 recruits GBAF to target genes in a BD-dependent manner**. BRD4 has also been shown to play a crucial role in maintaining ESC pluripotency by regulating the expression of several genes, including *Nanog*, *Lefty1*, and *Rex1*[38,39]. Indeed, we observed good correlation between genes from EpiESC-ESC dataset and DEGs from a published study comparing ESCs treated with either vehicle or JQ1, a small molecule that potently inhibits BRD4-BD binding to acetylated residues (common genes $n = 1666$) (Supplementary Figure 4a, $R^2 = 0.341$, linear regression)[39,40]. Similarly, gene expression following JQ1 treatment was highly correlated with I-BRD9 treatment at common DEGs ($n = 664$) (Fig. 7a, $R^2 = 0.646$, linear regression). This is not due to I-BRD9 nonspecifically targeting BRD4 and displacing it from chromatin because we observed no loss of BRD4 from the chromatin fraction with I-BRD9 using a cellular fractionation assay in ESCs (Supplementary Figures 4b, 4c). Likewise, JQ1 has no activity toward BRD9 in vitro[40]. These data suggest that BRD4 and the GBAF complex cooperate in regulating the naive pluripotency program.

This led us to ask whether BRD4 collaborates with the GBAF complex based on a physical interaction. We found that BRD4 and BRD9 engage in a transient interaction, as it is diminished in progressively harsher wash conditions (Supplementary Figure 4d). This explains why we did not observe peptides corresponding to BRD4 in our IP-mass spectrometry data, which was done under high stringency conditions. Additionally, only a small fraction of BRD4 co-sediments with GBAF complexes in the glycerol gradient sedimentation assay, confirming that it is not a bone fide GBAF subunit (Fig. 2b). We next investigated if BRD4 and GBAF co-localize on chromatin in ESCs. To this end, we performed ChIP-Seq against BRD4 in ESCs and observed 43,737 sites bound by BRD4 (Fig. 7b). Comparison with BRD9- and BRG1-binding sites revealed that 69% of BRD4-bound sites are co-bound by BRG1 while 47% are co-bound by BRD9, with 43% being bound by all three. On the other hand, we found that 25% of BRD4-bound sites are also bound by ARID1A (Supplementary Figure 4e). Altogether, our data suggest that an interaction between BRD4 and the GBAF complex accounts for their overlapping roles in regulating the naive pluripotency transcriptional network.

We next asked if the interaction between BRD4 and the GBAF complex is BD-dependent. We found that the interaction between BRD9 and BRD4 was enhanced by treating ESCs with the histone deacetylase I and II inhibitor trichostatin A (TSA) 6 h prior to nuclear lysate collection, which preserves acetylation on histones

and non-histone proteins (Fig. 7c, d). Addition of I-BRD9 reduced this interaction, indicating that the BRD9-BD potentially recognizes an acetylated form of BRD4. In light of this, we considered three possible modes of chromatin targeting—independent recruitment of BRD9 and BRD4 or targeting that is either BRD4-dependent or BRD9-dependent. To distinguish between them, we performed ChIP-Seq in ESCs against BRD4 with or without I-BRD9 and against BRD9 with or without JQ1. Consistent with our fractionation assay, I-BRD9 treatment had minimal effects on BRD4 chromatin targeting with only 492 sites being significantly decreased by 1.5-FC (Fig. 7e, Poisson $p$ value < 0.0001). This ruled out a BRD9-BD-dependent targeting of BRD4. On the other hand, JQ1 treatment resulted in 12,849 sites significantly losing BRD9 (Fig. 7f, Possion $p$ value < 0.0001), indicating that BRD4-BD is required for BRD9 localization on genomic targets. We compared the JQ1- and I-BRD9-sensitive BRD9 sites and found that there is substantial overlap between them, with 6965 common sites (Fig. 7g). In addition, when we compared the genes annotated to these common sites and the I-BRD9 DEGs, we found that essentially all of the genes associated with the naive pluripotency program lose BRD9 from chromatin in both JQ1 and I-BRD9 treatments (Fig. 7h). Altogether, these data point to a BRD4-BD-mediated recruitment of GBAF complexes to target sites that include key genes in the naive pluripotency network.

**BRD9 function is dispensable in 2i conditions**. Finally, it was recently shown that BRD4 is dispensable in the in vitro naive or ground pluripotent state, which is achieved by treating ESCs with glycogen synthase kinase 3β and MAP kinase kinase (MEK) inhibitors, commonly known as 2i[39]. We tested if this is also the case for BRD9 function and indeed, treatment of ESCs maintained in 2i conditions with I-BRD9 did not result in the dose-dependent decrease in cell growth that we observed in serum/LIF conditions (Fig. 8a). Consistent with this, I-BRD9-dependent reduction of *Nanog* and *Klf4* expression was blunted when ESCs were cultured in 2i conditions (Fig. 8b), similar to 2i-mediated rescue of pluripotency gene expression in JQ1-treated ESCs, as was previously shown[39]. Moreover, whereas NANOG protein expression was reduced by I-BRD9 in serum/LIF-cultured ESCs, it remained relatively high with the same treatment in 2i (Fig. 8c, d). Of note, BRD9 protein levels were decreased in both serum/LIF and 2i with either I-BRD9 and JQ1 treatment, likely due to it being targeted for degradation after being displaced from chromatin. Thus, BRD9 also appears to be non-essential in the ground state, in line with GBAF complexes and BRD4 functioning together in the same regulatory network.

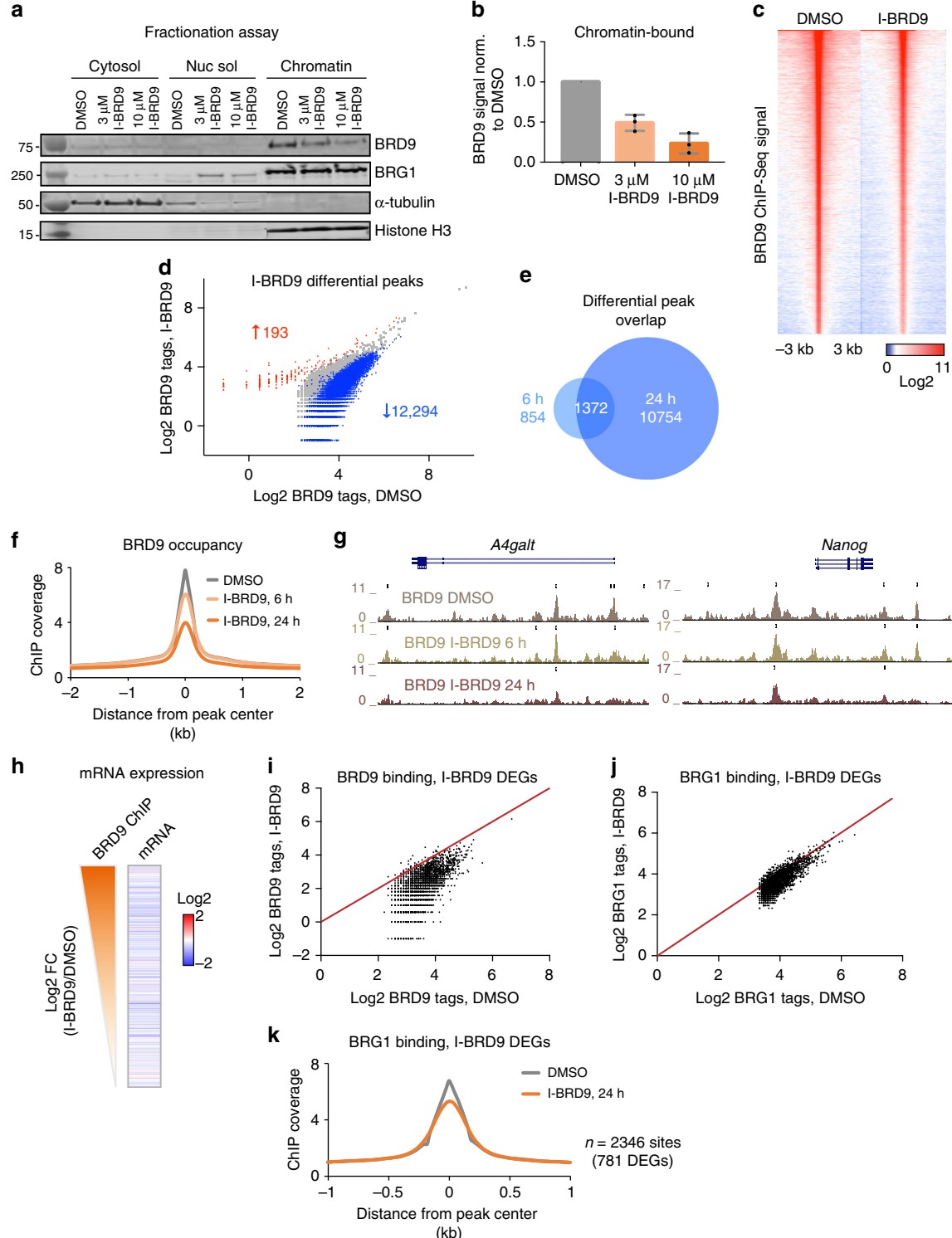

## Discussion

The mammalian BAF complex is highly polymorphic, assembling its subunits in a combinatorial manner that is cell type- or developmental stage-specific. Here we report that BRD9 associates with the non-canonical GBAF complex in ESCs that is distinct from esBAF and PBAF complexes. GBAF complexes are distinguished by the lack of BAF47, ARID1A, and BAF57, and the unique incorporation of GLTSCR1L or GLTSCR1. The lack of BAF47 is particularly notable due to its abilities to promote BRG1 binding to nucleosomal DNA and to stimulate BRG1's ATPase

and chromatin remodeling activities[41,42]. BAF47's absence suggests that BRD9 or another GBAF subunit can substitute for this function or that BRG1's enzymatic activity may be different in the context of GBAF. The function of GLTSCR1L or GLTSCR1 in GBAF complexes is not clear. Like other BAF subunits that have several paralogs, GLTSCR1 and GLTSCR1L are incorporated into mutually exclusive GBAF complexes[22]. In mouse ESCs, both paralogs are expressed, which likely explains why genetic deletion of GLTSCR1 in ESCs had no adverse effects on pluripotency[22]. Upon retinoic acid-induced differentiation[43], *Gltscr1* is

**Fig. 6** The bromodomain mediates BRD9 targeting to chromatin. **a** Representative immunoblotting analysis of a cellular fractionation assay in mouse ESC lysates after treatment with either DMSO or I-BRD9 at 3 or 10 μM for 24 h. Molecular weights from ladder are indicated. **b** Quantification of BRD9 chromatin fraction signal normalized to the loading control, Histone H3. Average of three independent experiments; error bars represent one standard deviation from the mean. Source data are provided as Source Data file. **c** Heat map of BRD9 ChIP signal ± 3 kb surrounding the BRD9 peak center in DMSO- or I-BRD9-treated ESCs, ranked according to BRD9 read density in DMSO. **d** Scatterplot of log2-transformed BRD9 ChIP tags in DMSO- and I-BRD9-treated ESCs. Blue and red correspond to 1.5-fold decrease or increase of BRD9 tag count in I-BRD9-treated ESCs, respectively (Poisson *p* value < 0.0001). **e** Venn diagram overlap of differentially decreased BRD9 ChIP sites after 6 and 24 h of I-BRD9 addition. **f** Histogram of BRD9 ChIP reads ± 2 kb surrounding the BRD9 peak center in DMSO and after 6 or 24 h of I-BRD9 treatment. **g** Representative genome browser tracks showing the progressive decrease in BRD9 ChIP reads with I-BRD9 treatment at 6 and 24-h time points. **h** Heat map of log2 FC in mRNA expression for genes annotated to BRD9 ChIP binding sites. Sites are ranked by degree of BRD9 ChIP signal loss after 24 h of I-BRD9 treatment, as indicated by an orange gradient. **i** Scatterplot of log2-transformed BRD9 ChIP tags in DMSO- and I-BRD9-treated ESCs at sites that are annotated to the genes that are significantly changed upon I-BRD9 treatment. Red line indicates *y* = *x* and corresponds to no change. **j** As in **i**, but for BRG1 ChIP tags in DMSO- and I-BRD9-treated ESCs. **k** Histogram of BRG1 ChIP reads ± 1 kb surrounding the BRG1 peak center in DMSO and after 24 h of I-BRD9 treatment, at sites annotated to I-BRD9 DEGs. Numbers of ChIP sites and of genes annotated to these sites are indicated

upregulated 3.6-fold while *Gltscr1l* is downregulated 3.7-fold, indicating a potential exchange between these mutually exclusive paralogs whereby GLTSCR1 becomes more dominant than GLTSCR1L in GBAF function during specific stages of development.

Our studies further establish that in addition to being biochemically distinct, GBAF and esBAF complexes are differentially targeted on the ESC genome. One remarkable difference is GBAF's localization at TAD boundaries and its strong enrichment at CTCF sites, which was also recently reported in cancer cell lines[24]. This indicates that GBAF could be playing a role in chromatin organization, which warrants future studies. The differential targeting on the genome also lends insight into how distinct BAF complexes are functionally integrated into the ESC pluripotency network. esBAF binding is enriched at sites bound by the general pluripotency regulators OCT4, SOX2, and NANOG, consistent with previous work demonstrating that BRG1 facilitates the binding of these factors to their target sites[12]. GBAF complexes, in contrast, support naive pluripotency by collaborating with KLF4 and Sp5. Specifically, genes downstream of the LIF/STAT3 pathway, including *Klf4* and its target *Nanog*, are downregulated in I-BRD9-treated cells, while genes involved in the pro-differentiation FGF/ERK pathway are upregulated. Previous reports have shown that BRG1 regulates the LIF/STAT3 pathway by maintaining accessibility at STAT3-binding sites[44]. While there is significant overlap between BRG1- and BRD9-dependent targets, our data specifically implicate KLF4 targets, suggesting that GBAF complexes play a functionally specific role in this pathway. Our studies thus distinguish functionally specific roles for BAF complex assemblies in maintaining ESC transcriptional programs.

Finally, we show that the BD of BRD9 is essential for targeting BRD9 to chromatin and affecting gene expression. In particular, at naive pluripotency gene targets, the BRD9-BD recognizes an acetylated form of BRD4, recruiting GBAF complexes to chromatin (Fig. 8e). BRD4 and BRD9 have been reported to co-localize at the *Myc* super enhancer in the mouse cell line model for acute myeloid leukemia, suggesting that recognition of acetylated BRD4 may be a common mechanism of GBAF complex recruitment[45]. Future studies are required to determine in which cell types the interaction occurs, which BRD4 acetylated residue is recognized by the BRD9-BD and what regulates this modification. Given the low-affinity interaction of a single BD with an acetylated lysine, it is likely that the interactions with other GBAF members, including a previously reported interaction between BRD4 and GLTSCR1[22,46], may stabilize the association between GBAF complexes and BRD4. Indeed, treatment with I-BRD9 does not completely inhibit the interaction between BRD4 and BRD9 (Fig. 7c, d). It is also important to note that

other mechanisms likely contribute to GBAF recruitment, for example BRD9-BD-dependent recognition of modifications on histone proteins or TFs. Indeed, a recent study from Evans and colleagues demonstrated that BRD9 is recruited via a BD-dependent interaction with lysine 91 on the Vitamin D receptor in islet cells[47]. Together, these studies demonstrate that reader domains in BAF complexes can serve as targeting modules for BAF complex recruitment. While BAF complexes contain many such domains, none of these domains has been shown to affect BAF complex targeting, which is primarily mediated through TFs[48–50]. It is assumed that the multivalent binding from multiple domains ensures appropriate targeting and buffers BAF complexes against the effects of single domain inhibition. It is thus significant that BRD9-BD inhibition leads a rapid loss of BRD9 from chromatin, demonstrating that BAF complexes can be recruited or stabilized by interactions with chromatin.

In summary, our studies provide compelling evidence that BRD9 functions within the naive pluripotency regulatory network by associating with a non-canonical GBAF complex, which is recruited to target sites via BD-dependent recognition of BRD4. This further expands the function of BAF complexes in stem cell biology and contributes to our understanding of how biochemical diversity in BAF complex assembly provides increased regulatory control in transcriptional programs.

## Methods

**Cell culture**. v6.5 mouse ESCs (Salk Institute Transgenic Core) and were *Arid1a*^f/f^; ActinCreERT2 ESCs[51,52] were cultured in Knockout™ Dulbecco's modified Eagle's medium (Thermo Fisher Sci #10829018) supplemented with 15% ES-qualified serum (Applied Stem Cell Inc. #ASM-5007) and Knockout™ Serum Replacement (Thermo Fisher Sci #10828028), 2 mM L-glutamine (Gibco #35050061), 10 mM HEPES (Gibco #15630080), 1 mM sodium pyruvate (Gibco #11360070), 100 U mL⁻¹ penicillin/streptomycin (Gibco #15140122), 0.1 mM non-essential amino acids (Gibco #11140050), 0.1 mM beta-mercaptoethanol (Gibco #21985023), and LIF. ESCs were maintained on gamma-irradiated mouse embryonic fibroblast (MEF) feeders for passage or gelatin-coated dishes for assays at 37 °C, 5% CO₂ with daily media changes and passaged every other day. For ESCs cultured in 2i, media as described above were supplemented with 3 μM CHIR99021 (LC Laboratories C556) and 1 μM PD0325901 (BioTang 391210–10–9) and cells were grown on gelatin-coated dishes. *Arid1a*^f/f^;ActinCreERT2 ESCs were treated with either vehicle or 1 μM 4-hydroxytamoxifen (Sigma, dissolved in ethanol) for up to 24 h to induce *Arid1a* deletion.

HCT116 colorectal cancer cells were purchased from Horizon Discovery (American Type Culture Collection: CCL-247) and cultured in RPMI (Life Technologies #11875–085) supplemented with 10% fetal bovine serum (Omega Sci FB-11 Lot #419414) and 100 U mL⁻¹ penicillin/streptomycin (Gibco #15140122).

Stocks of the following small-molecule inhibitors were made in dimethyl sulfoxide (DMSO): I-BRD9 (Cayman Chemicals #17749), BI-9564 (Cayman Chemicals #17897), TP472 (Tocris #6000), (+/−) -JQ1 (Sigma #SML0974), PFI-3 (Sigma #SML0939), and TSA (Sigma T8552). For phenotypic assays, ESCs were treated with increasing concentrations of inhibitors. For IP experiments, ESCs were treated with either vehicle or 200 nM TSA for 6 h prior to nuclear lysate collection.

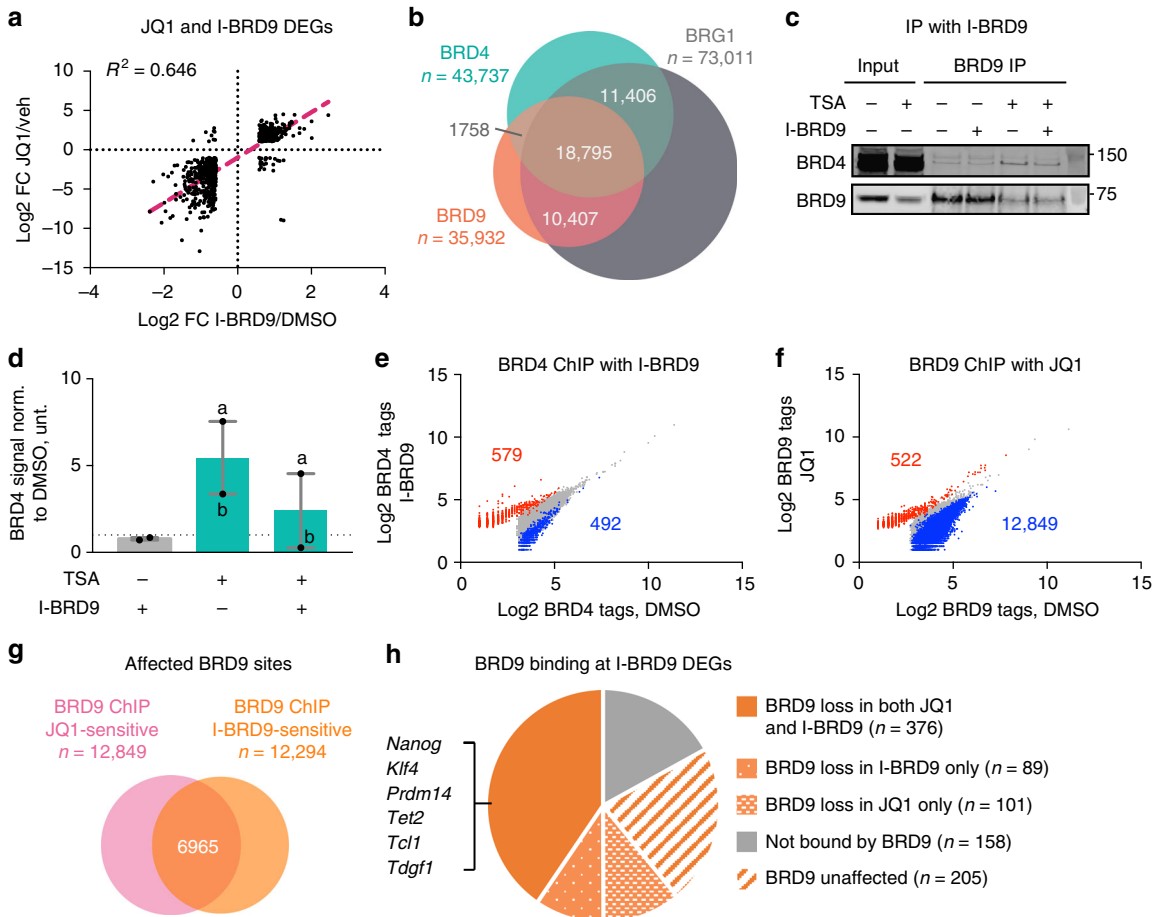

**Fig. 7** BRD4 recruits GBAF to target gene sites in a BD-dependent manner. **a** Scatterplot of the mRNA log2 FCs in I-BRD9/DMSO and JQ1/vehicle for 664 common DEGs. Linear regression analysis was performed to calculate the $R^2$, with the best fit shown as a pink dashed line. **b** Venn diagram of overlaps between BRG1, BRD9, and BRD4 ChIP binding sites in ESCs, with $n$ representing the number of observed peaks. **c** ESCs were treated with either DMSO or 200 nM of HDAC inhibitor trichostatin A (TSA) for 6 h prior to nuclear lysate collection then IP experiments were performed against BRD9 with or without I-BRD9. Molecular weights from ladder are indicated. **d** Quantification of BRD4 signal normalized to BRD9 bait signal then normalized to untreated from two biological experiments labeled a and b. Source data are provided as Source Data file. **e** Scatterplot of log2-transformed BRD4 ChIP tags in DMSO- and I-BRD9-treated ESCs. Blue and red correspond to 1.5-fold decrease or increase of BRD4 tag count in I-BRD9-treated ESCs, respectively (Poisson $p$ value < 0.0001). **f** As in **e**, but for BRD9 ChIP tags in DMSO- and JQ1-treated ESCs. **g** Venn diagram of the overlap between JQ1-sensitive and I-BRD9-sensitive BRD9 ChIP sites. **h** Pie chart showing the number of I-BRD9 DEGs that are BRD9-bound in DMSO and those that significantly lose BRD9 binding upon treatment with I-BRD9, JQ1, or both (FC 1.5, Poisson $p$ value < 0.0001)

**List of shRNAs**. *Brd9* #1: TTTATTTCTTCTTTCATCTTTG (Addgene #75130). The hairpin was restriction enzyme digested from the retroviral vector with *Mlu*I and *Xho*I and ligated into pGipZ lentiviral vector (NEB Quick Ligation Kit). Non-targeting hairpin in pGipZ was used as a control. *Brd9* #2: ATCAGGCT CAGGTGCGTTC (Dharmacon V3SM1241–231713739) and *Brd9* #3: TTGAGTGATCACCACCTGT (Dharmacon V3SM1241–233485164) were used as provided. Non-targeting hairpin (Dharmacon VSC11709) was used as a control.

**Lentivirus preparation and ESC infection**. HEK293T cells were transfected with the lentiviral constructs and packaging plasmids Md2G and psPAX2 using polyethylenimine-mediated transfection. Forty-eight hours post transfection, the media containing the virus was collected, filtered, and centrifuged at 70,952 × *g* for 2 h at 4 °C. The viral pellet was resuspended in 1× phospho-buffered saline (PBS) and stored at −80 °C until use. v6.5 mouse ESCs were infected in suspension with the concentrated virus for 1–2 h with 5 µg mL$^{-1}$ of polybrene then plated onto MEF feeders for incubation with the virus overnight. Media was changed the next day. Twenty-four hours later, shRNA-expressing cells were selected with 1 µg mL$^{-1}$ of puromycin on puromycin-resistant MEFs for 48 h.

**Cell growth assay**. For hairpin-transduced v6.5 mouse ESCs, 100K cells per well were plated on a gelatin-coated 12-well plate 2 days after puromycin selection. Cells were counted using Trypan Blue exclusion method on the TC20 Cell Counter (Biorad) after 2 or 4 days. Cells were passaged at day 2 to inhibit contact-induced differentiation.

For the small-molecule inhibitor experiments, v6.5 mouse ESCs were plated at 100K cells per well on a gelatin-coated 12-well plate. Twenty-four hours later, the cells were treated with either vehicle (DMSO) or varying concentrations of the following inhibitors: I-BRD9, BI-9564, PFI-3, or TP472. Cells were counted using Trypan Blue exclusion method on the TC20 Cell Counter (Biorad) every 48 h, during which the cells were re-plated at the same cell density on a new 12-well plate.

Two-tailed *t*-test was performed to obtain the *p* values from biological replicates, $n = 9$ for I-BRD9, $n = 6$ for BI-9564 and TP472, $n = 3$ for PFI-3, and $n = 3$ for each sh*Brd9* hairpin. *$p$ < 0.05, **$p$ < 0.01, ***$p$ < 0.001.

**Clonogenicity assay**. Cells were seeded at 100 cells per 9 cm$^2$ in gelatin-coated six-well plates. Cells were treated continuously from day of plating with either vehicle (DMSO) or the indicated concentrations of BRD9 inhibitors, I-BRD9 or BI-9564. Media was replaced every day and at day eight, colonies were counted.

For detection of AP activity, cells were seeded and treated as indicated above. At day 8, cells were fixed with 4% paraformaldehyde in 1× PBS, rinsed with Tris-buffered saline solution (TBS) with 0.05% Tween-20 and stained using the Alkaline Phosphatase Detection Kit (Millipore Sigma SCR004) per the manufacturer's instructions.

**Density sedimentation assay**. v6.5 mouse ESCs or HCT116 cells were lysed in Buffer A (25 mM HEPES, pH 7.6, 5 mM MgCl$_2$, 25 mM KCl, 0.05 mM EDTA, 10% glycerol, and 0.1% NP-40) supplemented with 1 mM dithiothreitol (DTT), 1 mM

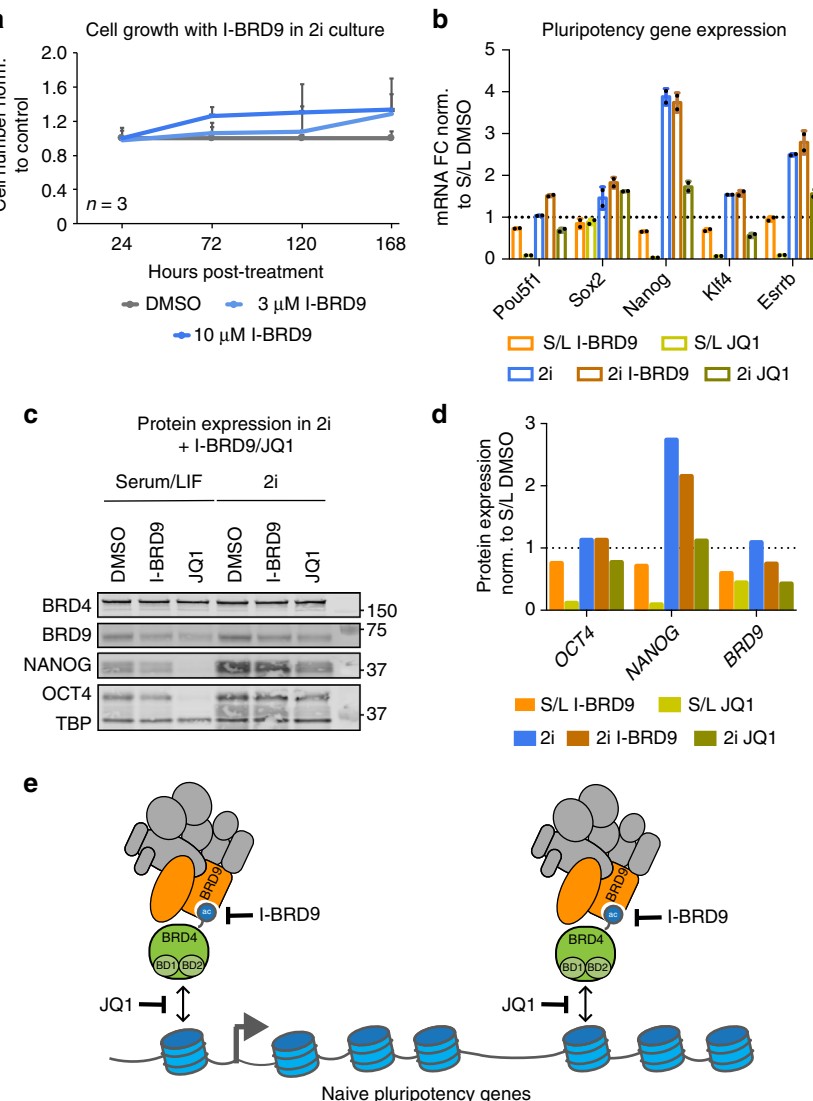

**Fig. 8** BRD9 is dispensable for regulating the naive pluripotency network in 2i. **a** Time course experiment assessing mouse ES cell number cultured in 2i conditions (MEK inhibitor PD0325901 and GSK3 inhibitor CHIR99021), after treatment with DMSO or I-BRD9 at either 3 or 10 μM. Error bars represent one standard deviation from the mean of three biological replicates. **b** Bar graph showing the fold change in pluripotency gene expression after 72 h of treatment with I-BRD9 or JQ1 of ESCs cultured in serum/LIF (S/L) or 2i. Mean of two technical replicates; error bars represent one standard deviation from the mean. Source data are provided as a Source Data file. **c** Immunoblotting analysis of ESC nuclear lysates after 72 h of treatment with I-BRD9 or JQ1 of ESCs cultured in S/L or 2i. Molecular weights from ladders are indicated. Source data are provided as a Source Data file. **d** Quantification of the indicated proteins in **c**, normalized to TATA-binding protein (TBP) loading control then normalized to values in S/L DMSO. Dotted line is $y = 1$, which represents S/L DMSO. **e** Model of BRD4-mediated recruitment of BRD9/GBAF to sites annotated to naive pluripotency genes

phenylmethylsulfonyl fluoride (PMSF), 1 μM pepstatin, 10 μM leupeptin, and 10 μM chymostatin at $10^7$ cells per 5 mL and incubated on ice for 10–15 min. Nuclei were spun down at 900 xg for five minutes then resuspended in Buffer C (10 mM HEPES, pH 7.6, 3 mM MgCl₂, 100 mM KCl, 0.1 mM EDTA, and 10% glycerol) supplemented with 1 mM DTT, 1 mM PMSF, 1 μM pepstatin, 10 μM leupeptin, and 10 μM chymostatin at $40 \times 10^6$ cells per 945 μL. Ammonium sulfate was added at a final concentration of 300 mM, incubated on an end-to-end rocker in the cold room for 30 min, then spun down at $446,082 \times g$ for 10 min. Nuclear proteins were precipitated by incubation with ammonium sulfate at a final concentration of 0.3 mg mL$^{-1}$, on ice for 20 min then centrifugation at $446,082 \times g$ for 10 min. Dry pellet was stored at −80 °C until use.

Seven hundred micrograms of nuclear proteins were resuspended in 150 μL of HEMG solution (25 mM HEPES, pH 7.9, 0.1 mM EDTA, 12.5 mM MgCl₂, and 100 mM KCl) supplemented with 1 mM DTT, 1 mM PMSF, 1 μM pepstatin, 10 μM leupeptin, and 10 μM chymostatin then overlaid onto 10 mL of HEMG solution with 10–30% glycerol gradient prepared in a 14 × 89 mm polyallomer centrifuge tube (Beckman). Proteins were subjected to ultra-centrifugation in a SW40 rotor at 4 °C for 16 h at $283,807 \times g$. The next day, 0.5 mL fractions were collected and analyzed using immunoblotting (IB).

**List of antibodies.** ARID1A: Santra Cruz sc-32761 or Millipore 04-080 (IB 1:1000 v/v and ChIP 10 μL for $6.25 \times 10^6$ cells)

BRG1: Santa Cruz sc-17796 (IB 1:1000 v/v) or Abcam 110641 (IP 1:100 m/m, IB 1:2000 v/v, and ChIP 5 μL for $6.25 \times 10^6$ cells)

BAF155: Santa Cruz sc-10756 (IB 1:1000 v/v) or in-house antibody[19] (IP 1:100 m/m)

PBRM1: Bethyl A301–591A, 1:2000 v/v

GLTSCR1: Santa Cruz sc-515086, 1:1000 v/v

GLTSCR1L: Thermo Fisher Sci PA5–56126, 1:500 v/v

BRD9: Active Motif 61537 (IP 1:100 m/m, IB 1:2000 v/v, and ChIP 5 μL for $6.25 \times 10^6$ cells)

BRD7: Santa Cruz sc-376180, 1:500 v/v

BRD4: Bethyl A301–985A50 (IP 1:100 m/m, IB 1:2000 v/v, and ChIP 5 μL for $6.25 \times 10^6$ cells)

BAF47: Santa Cruz sc-166165 (IP 1:100 m/m and IB 1:1000 v/v)

BAF57: Bethyl A300–810A, 1:2000 v/v

BAF60A: Santa Cruz sc-135843, 1:1000 v/v

BAF53A: Novus Bio NB100–61628, 1:2000 v/v

IgG: rabbit, Santa Cruz sc-2027 (IP-mass spec 5 μL for 1 mg of nuclear lysates) or Cell Signaling 2729S (IP 1:100 m/m)

TBP: Thermo Fisher MA1–189, 1:2000 v/v
OCT4: Santa Cruz sc-5279, 1:1000 v/v
NANOG: Abcam ab80892, 1:1000 v/v

**Immunoprecipitation**. Nuclear lysates were collected from v6.5 mouse ESCs or HCT116 cells following a revised Dignam protocol[53]. After cellular swelling in Buffer A (10 mM HEPES, pH 7.9, 1.5 mM MgCl$_2$, and 10 mM KCl) supplemented with 1 mM DTT, 1 mM PMSF, 1 μM pepstatin, 10 μM leupeptin, and 10 μM chymostatin, cells were lysed by homogenization using a 21-gauge needle with six to eight strokes. If lysis remained incomplete, cells were treated with 0.05–0.1% NP-40 for 10 min on ice prior to nuclei collection. Nuclei were spun down at 900 × g for 5 min then resuspended in Buffer C (20 mM HEPES, pH 7.9, 20% glycerol, 420 mM NaCl, 1.5 mM MgCl$_2$, and 0.2 mM EDTA) supplemented with 1 mM DTT, 1 mM PMSF, 1 μM pepstatin, 10 μM leupeptin, and 10 μM chymostatin. After 30 min of end-to-end rotation in the cold room, sample was clarified at 21,100 × g for 10 min. Supernatant was collected and flash-frozen in liquid nitrogen, if necessary.

Prior to the IP, the nuclear lysates were diluted with two-thirds of original volume of 20 mM HEPES, pH 7.9, and 0.3% NP-40 to bring down the NaCl concentration. A unit of 200–300 μg of nuclear lysates was used per IP with antibodies against BRG1, BRD9, or BAF47 overnight at 4 °C. Precipitated proteins were bound to 50:50 Protein A, Protein G Dynabeads (Invitrogen) for 1–2 h and washed with either RIPA (50 mM Tris, pH 8, 150 mM NaCl, 1% NP-40, 0.5% sodium deoxycholate, and 0.1% SDS) for BRD9IP or Wash Buffer (50 mM Tris, pH 8, 150 mM NaCl, 1 mM EDTA, 10% glycerol, and 0.5% Triton X-100) for BRG1- and BAF47-IPs. Proteins were eluted in SDS-polyacrylamide gel electrophoresis (SDS-PAGE) loading solution with boiling and analyzed by IB.

For IP with BRD9 with or without I-BRD9, nuclear lysates were split into equal volume and incubated with or without 10 μM I-BRD9 (diluted fresh in binding buffer) for 30 min prior to addition of antibody. For GLTSCR1/GLTSCR1L/BAF60A blotting, the IP was washed with 50 mM Tris, pH 8, 150 mM NaCl, 1 mM EDTA, 10% glycerol, and 0.5% Triton X-100. For BRD4 blotting, the IP was washed with 50 mM Tris, pH 8, 150 mM NaCl, and 0.1% NP-40. I-BRD9-treated samples were washed with buffers that contained 20 μM I-BRD9.

For BRG1 and BAF155 depletion assay, nuclear lysates were subjected to four rounds of incubation with the respective antibodies and Protein A + G dynabeads, minimum of 3 h to overnight at 4 °C. The supernatant flowthrough from each IP was collected and analyzed by IB.

**Urea-based denaturation assay**. v6.5 mouse ESC nuclear lysates were collected as described above in the Immunoprecipitation section. Two hundred micrograms of the nuclear lysates were subjected to varying concentrations of urea (0.25–5 M) in 25 mM Tris, pH 8, 150 mM NaCl, 0.1% NP-40, and 1 mM DTT for 15 min at room temperature (RT) prior to performing the IP with an antibody against BRD9. The precipitated proteins were washed and eluted as described above and analyzed by IB.

**IB assay**. Protein samples were run on 4–12% Bis-Tris gels (Life Technologies). After primary antibody incubation, blots were probed with 1:20,000 v/v dilution of either fluorescently labeled secondary antibodies (Life Technologies #A21058, Invitrogen #SA535571) in 2% bovine serum albumin in PBST or horseradish peroxidase-conjugated anti-rabbit secondary antibody (Veriblot Abcam #131366) in 5% non-fat milk in TBST for an hour at RT. Fluorescent images were developed using Odyssey. Veriblot-probed blots were treated with enhanced chemiluminescence substrate (Biorad #170–5060) for 5 min then developed on film. Original scans of all blots are included as a Source Data File.

**Cellular fractionation**. Fractionation of v6.5 ESCs treated with either DMSO or I-BRD9 at 3 or 10 μM for 24 h was done according to published protocol[54]. Briefly, 20 × 10$^6$ cells were lysed in Buffer 1 (10 mM HEPES, pH 7.9, 10 mM KCl, 1.5 mM MgCl$_2$, 0.34 M sucrose, 10% glycerol, 1 mM DTT, protease inhibitors, and 0.1% Triton X-100). After 8 min on ice, nuclei were harvested by centrifugation at 1,300 × g for 5 min then resuspended in Buffer 2 (3 mM EDTA, 0.2 mM EGTA, 1 mM DTT, and protease inhibitors). Supernatant 1 is the cytosolic fraction. Samples were spun at 20,000 × g to isolate chromatin fraction (pellet), which was subsequently resuspended in 100 μL of 1× SDS-PAGE loading dye in TBS + 100 mM DTT and incubated at 70 °C for 10 min. Supernatant 2 is the nuclear soluble fraction. For loading onto SDS-PAGE gels, sample viscosity was reduced by dilution.

**IP-mass spectrometry**. Rabbit polyclonal IgG and BRD9-specific antibodies were crosslinked to Dynabeads (Invitrogen) using bis(sulfosuccinimidyl) suberate (BS3). Briefly, Dynabeads were blocked by incubating with 10 μg μL$^{-1}$ sheared, salmon-sperm DNA in wash buffer (WB) (0.1 M NaPO4, pH 8.2, and 0.1% Tween-20) then incubated with antibody at RT for 15 min. After two washes in conjugation buffer (20 mM NaPO4, pH 8.2, and 150 mM NaCl), the antibody-beads complexes were incubated with 5 mM BS3 for 30 min at RT. Crosslinking was quenched with Tris-HCl, pH 7.4, and the antibody-beads complexes were washed with conjugation

buffer and equilibrated with IP buffer (20 mM Tris, pH 8, 150 mM NaCl, and 0.1% NP-40).

IP was performed from 1 mg of v6.5 mouse ESC nuclear lysates with either rabbit IgG or BRD9-specific antibody. Precipitated proteins were treated with Micrococcal nuclease S7 for 15 min, washed with RIPA buffer, and eluted in 20 mM Tris, pH 8, 150 mM NaCl, 1× SDS-PAGE loading dye, and 1 mM DTT with boiling. Samples were precipitated by methanol/chloroform. Dried pellets were dissolved in 8 M urea/100 mM TEAB, pH 8.5. Proteins were reduced with 5 mM tris(2-carboxyethyl)phosphine hydrochloride (Sigma-Aldrich) and alkylated with 10 mM chloroacetamide (Sigma-Aldrich). Proteins were digested overnight at 37 °C in 2 M urea/100 mM TEAB, pH 8.5, with trypsin (Promega). Digestion was quenched with formic acid, 5% final concentration.

The digested samples were analyzed on a Fusion Orbitrap tribrid mass spectrometer (Thermo) in a data-dependent mode.

Protein and peptide identification was done with PatternLab for Proteomics[55]. Tandem mass spectra were searched with Comet[56] against a mouse UniProt database. The search space included all fully tryptic and half-tryptic peptide candidates. Carbamidomethylation on cysteine was considered as a static modification. Data were searched with 40 ppm precursor ion tolerance. Identified proteins were filtered using SEPro[55] and utilized a target-decoy database search strategy to control the false discovery rate to 1% at the protein level[57].

**RNA-Seq sample preparation**. v6.5 mouse ESCs were transduced in suspension with either pooled sh*Brd9* or shcontrol lentivirus for 1–2 h then plated onto MEF feeders for incubation with virus overnight. RNA was collected 4 days post puromycin (1 μg mL$^{-1}$) selection on puro-resistant MEF feeders. *Arid1a*$^{f/f}$:CreERT2 ESCs were cultured on gelatin-coated dishes, treated with either ethanol or 1 μM tamoxifen then passaged 24 h post treatment. Forty-eight hours after passage, RNA was collected. v6.5 mouse ESCs treated with either DMSO or 10 μM I-BRD9 were cultured on MEF feeders then passaged onto gelatin-coated dishes 24 h prior to RNA collection.

RNA from 1–3 × 10$^6$ cells was extracted and purified with the Zymo Research Quick-RNA miniprep kit according to the manufacturer's instructions. RNA-Seq libraries were prepared using Illumina TruSeq Stranded mRNA kit following the manufacturer's instructions with 5 μg of input RNA.

**Quantification of gene expression**. Seventy-two hours after v6.5 mouse ESCs were treated with DMSO, 10 μM I-BRD9, or 500 nM JQ1, RNA samples were isolated using Quick-RNA Miniprep Kit (Zymo Research). cDNA synthesis was performed with 2 μg of RNA using SuperScript III and oligo-dT primer in 20 μL reaction volume per the manufacturer's protocol (Invitrogen #18080–051). Two microliters of 1:50 diluted cDNA samples (in water) were used per reaction. Quantitative real-time PCR analysis was performed in technical duplicates using CFX Real Time System (Biorad) with iTaq Universal SYBR Green Supermix (Biorad #64163963). *Gapdh* was used as the endogenous control. The sequences of primers used are listed in Supplementary Table 2.

**ChIP-Seq sample preparation**. Approximately, 20 × 10$^6$ v6.5 mouse ESCs cultured on gelatin treated with DMSO or 3 μM I-BRD9 for 6 or 24 h or 500 nM JQ1 for 24 h were collected and crosslinked first in 3 mM disuccinimidyl glutarate in 1× PBS then in 1% formaldehyde. After quenching the excess formaldehyde with 125 mM glycine, the fixed cells were washed, pelleted, and flash-frozen. Upon thawing, the cells were resuspended in lysis solution (50 mM HEPES-KOH, pH 8, 140 mM NaCl, 1 mM EDTA, 10% glycerol, 0.5% NP-40, and 0.25% Triton X-100, and incubated on ice for 10 min. The isolated nuclei were washed with wash solution (10 mM Tris-HCl, pH 8, 1 mM EDTA, 0.5 mM EGTA, and 200 mM NaCl) and shearing buffer (0.1% SDS, 1 mM EDTA, and 10 mM Tris-HCl, pH 8) then sheared in a Covaris E229 sonicator for 10–20 min to generate DNA fragments between ~200 and 1000 base pairs (bp). After clarification of insoluble material by centrifugation, the chromatin was immunoprecipitated overnight at 4 °C with antibodies against BRG1, BRD9, and ARID1A bound to Protein A + G Dynabeads (Invitrogen) in ChIP buffer (50 mM HEPES-KOH, pH 7.5, 300 mM NaCl, 1 mM EDTA, 1% Triton X-100, 0.1% DOC, and 0.1% SDS). Antibody-bound DNA was washed and treated with Proteinase K and RNase A and crosslinking was reversed by incubation at 55 °C for two and a half hours. Purified ChIP DNA was used for library generation (NuGen Ovation Ultralow Library System V2) according to the manufacturer's instructions for subsequent sequencing.

**RNA-Seq analysis**. Single-end 50 bp reads were aligned to mm10 using STAR alignment tool (V2.5)[58]. RNA expression was quantified as raw integer counts using analyzeRepeats.pl in HOMER using the following parameters: -strand both -count exons -condenseGenes –noadj. To identify DEGs, we performed getDiffExpression.pl in HOMER, which uses the DESeq2 R package to calculate the biological variation within replicates. Cut-offs were set at log2 FC = 0.585 and FDR at 0.05 (Benjamin-Hochberg). For RNA expression of nearest annotated gene for sites that lose BRD9 ChIP binding with I-BRD9, ChIP peaks were annotated to the closest transcription start site (TSS) and the associated log2 fold change (I-BRD9/DMSO) was determined.

**GO analysis**. GO analysis was performed on the list of 929 I-BRD9 DEGs on the GSEA website (GSEA homepage, [www.gsea-msigdb.org/], 2004–2017).

**Gene set enrichment analysis**. GSEA software was used to perform the analyses with the following parameters: number of permutations = 1000; enrichment statistic = weighted; and metric for ranking of genes = difference of classes (input RNA-Seq data were log-transformed).

**ChIP-Seq analysis**. Single-end 50 bp reads were aligned to mm10 using STAR alignment tool (V2.5)[58]. ChIP-Seq peaks were called using findPeaks within HOMER using default parameters for histone (-style histone) or TF (-style factor). Peaks were called when enriched greater than twofold over input and greater than fourfold over local tag counts, with FDR 0.001 (Benjamin-Hochberg). For histone ChIP, peaks within a 1000 bp range were stitched together to form regions. ChIP-Seq peaks or regions were annotated by mapping to the nearest TSS using the annotatePeaks.pl command. Differential ChIP peaks were found by merging peaks from control and experiment groups and called using getDiffExpression.pl with fold change ≥ 1.5 or <−1.5, Poisson $p$ value < 0.0001. For peak calling with replicate samples, we used the getDifferentialPeaksReplicates.pl program, with -style factor and default parameters for FC and Poisson $p$ value.

Significance of peak overlap was determined by calculating the number of peaks co-occurring across the entire genome using the HOMER mergePeaks program. For enhancer enrichment analysis, we defined the enhancer classes using publicly available mouse ESC ChIP-seq data for Mediator and histone modifications (see Data availability)[27,59,60]. Enhancers were called by identifying all H3K4me-positive regions that are at least 1 kb away from the nearest TSS or H3K4me3 mark. These were subdivided as active (H3K27ac-positive) or poised (H3K27ac-negative)[60]. We then differentiated the H3K4me-positive and H3K27ac-positive regions into active and super enhancers by ranking the regions by Mediator ChIP-Seq tag density and using the tangent of the curve to call super enhancers[27].

Promoter annotation was performed using the HOMER annotatePeaks program, which by default is −1 kb to +100 bp away from a known TSS. Distal sites were called using the HOMER getDistalPeaks program, which finds intergenic regions but excludes transcription termination sites.

**Motif analysis**. Sequences within 200 bp of peak centers were compared to known motifs in the HOMER database using the findMotifsGenome.pl command with the following fragment size and motif length parameters, respectively: -size 200 -len 8. Random GC content-matched genomic regions were used as background (default). Enriched motifs are statistically significant motifs in input over background by a $p$ value of <0.05. $p$ Values were calculated using cumulative binomial distribution.

**TAD boundary enrichment analysis**. Hi-C data from mESCs[61] were downloaded from the GEO database (GSE96107) and mapped to the mm10 genome using bwa-mem. Reads were paired manually using an in-house pipeline, and PCR duplicate reads were removed using Picard. TADs were called using the directionality index (DI) method[28]. Briefly, Hi-C interaction matrices were converted to a vector of upstream and downstream interaction frequency bias using a chi-squared liked statistic, termed the DI. The DI values were then used as input for a Gaussian mixture hidden Markov model to identify local states of upstream and downstream bias in interaction frequencies. Domains were called as regions of continuous downstream biased states and ends when the last in a series of upstream biased states are reached. The regions between the topological domains are termed TAD boundaries if they are <400 kb or unorganized chromatin if they are more than 400 kb. Enrichment of ChIP-Seq data over TADs was calculated by partitioning each TAD into 100 bins, and also considering 50 bins upstream and downstream of the domain. The number of peaks per kb per bin was calculated and averaged across all domains in the genome. To compare across samples with different number of peaks, the final averaged values were normalized by the number of peaks in each dataset divided by 10,000.

**Statistical tests**. Statistically significant differences in cell growth assays: Two-tailed $t$ tests were performed to calculate $p$ values on Graphpad Prism Version 7. Number of replicates are provided above, in the Cell growth assay section.

Overlap between datasets: $p$ values were calculated using hypergeometric test of overlap with population size being the total number of genes tested ($N = 24,538$), using an online tool found at http://www.rothsteinlab.com/tools/apps/hyper_geometric_calculator.

Correlation of RNA-Seq values between datasets: Goodness of fit ($R^2$) was analyzed using linear regression on Graphpad Prism Version 7.

## Data availability

The raw mass spectrometry files for BRD9-interacting proteins from mESCs have been deposited into the ProteomeXchange Consortium via the PRIDE partner repository with the following identifier PXD010670. RNA-seq and ChIP-Seq data that support the findings of this study have been deposited in the Gene Expression Omnibus under the accession code GSE111264. We also used publicly available sequencing data, which were processed using HOMER v4.8 (Christopher Benner,

HOMER, http://homer.ucsd.edu/homer/index.html, 2018): histone modifications H3K4me, H3K4me3, and H3K27ac (GEO GSE24165); H3K27me3 (GEO GSM1397343); Mediator (GEO GSE44288); CTCF (GEO GSE30203); Sp5 (GEO GSE72989); KLF4 (GEO GSM288354); OCT4, NANOG, and SOX2 (GEO GSE44286); c-Myc (GEO GSM288356); RNA-Seq Brg1$^{f/f}$ (GEO GSE87820); RNA-Seq ESC-EpiESC (GEO GSE79796); RNA-Seq JQ1 (GEO GSE88760); RNA-Seq RA-induced differentiation (GSE39522); and Hi-C data (GSE96107). Source data are provided for Figs. 1a–c, 2, 5a–e, 6a, 7c, 0d, 8a–c, Supplementary Figures 1, 2, 3 and 4b-d as a Source Data file. All other data are available from the corresponding author upon reasonable request. A Reporting Summary for this Article is available as a Supplementary Information file.

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

## Acknowledgements

Sequencing was carried out by the NGS Core Facility of the Salk Institute with funding from NIH-NCI CCSG: P30 014195, the Chapman Foundation and the Helmsley Charitable Trust. We thank N. Hah and T. Nguyen for technical support. The Razavi Newman Integrative Genomics and Bioinformatics Core Facility of the Salk Institute is funded through the NIH-NCI CCSG: P30 014195 and the Helmsley Trust. The IP-mass spectrometry sample preparation and LC/MS collection was carried out by the Mass Spectrometry Core of the Salk Institute with funding from NIH-NCI CCSG: P30 014195 and the Helmsley Center for Genomic Medicine. We thank J Moresco and J Diedrich for technical support. We are grateful to Drs. Ron Evans and Tony Hunter (Salk Institute for Biological Studies) for critical reading of the manuscript and to the members of the Hargreaves Lab for their valuable feedback. D.C.H. is supported by the NIH R00 CA184043-03 and the V Foundation for Cancer Research V2016-006. J.G. is supported by the Salk Institute T32 Cancer Training Grant T32CA009370 and the NIGMS NRSA F32 GM128377-01.

## Author contributions

J.G. and D.C.H. designed the experiments. J.G., S.M., J.H., and D.C.H. performed the experiments and J.G. and D.H. analyzed the data. J.G. and T.W.R.K. performed the bioinformatics analyses under the supervision of D.C.H. D.-S.L. and J.R.D. performed the Hi-C analysis. M.S. assisted with the bioinformatics analyses. J.G. and D.C.H. wrote and edited the manuscript.

## Additional information

**Competing interests:** The authors declare no competing interests.

