## [Peer Review File · Nature Communications]

Reviewers' comments:

Reviewer #1 (Remarks to the Author):

Review of Nature Communications NCOMMS-18-05631 by Gatchalian et al., (Hargreaves Lab)

This work addresses issues of high interest in the fields of chromatin remodeling complexes; the composition and functions of particular sub-complexes of the BAF chromatin remodeler, as this complex plays such central roles in development and cancer. Here the authors describe a sub-complex with BRD9, termed BBAF, that lacks several components normally associated with BAF/PBAF, and examine the function of BRD9 and BBAF, as well as localization in the genome of ES cells. The most novel and interesting observation is the different locations of the esBAF complex and the BBAF complex, and the correlations with particular transcription factors.

Overall, the work is well designed and executed. The work begins with BRD9 inhibitors and shRNAs to show the impact on ESC growth and transcription, revealing an impact significantly focused on developmental genes. Gel filtration of ESC nuclear extracts reveals a lack of co-elution of BRD9 with ARID1a and certain other BAF components (including BAF47) in a smaller complex termed BBAF. Co-IP analysis confirms the composition (for the components tested). Furthermore, IP/mass spec reveals the full composition. Overall, I found this work convincing and interesting. It is, however, no longer novel, as recent work from Alpsy and Dykhuizen (JBC 2018) have recently published the existence and composition of this complex, which is identical to their 'GBAF' complex.

Next, the authors perform ChIP on ARID1A, BRG1 and BRD9, and get an interesting set of results. Consistent with separate complexes, there are more unique binding sites than common binding sites, as well as a difference in preference for promoters versus distal sites – especially the co-incidence of the BBAF complex at H3K4me3-marked promoters. Next, analysis of the co-occupied transcription factors reveals co-incidence of BBAF specifically with CTCF, and esBAF with core pluripotency factors. I found this quite interesting.

The next section examines BRD9 inhibition and ARID1A mutants in terms of pluripotency, with the data supporting states of primed ESCs in the absence of BRD9, suggesting that BRD9 protects naïve pluripotent states. Correlations are also found between Klf4 and BBAF binding and gene regulation. Finally, chromatin localization is shown to rely on BRD9 at many sites, suggesting a role in either targeting or retention.

Overall, I find this a really interesting paper. The novelty of the first section is diminished by the recent JBC paper, but the middle and end of the paper involving the genomics analyses are novel and interesting, and provide several pieces of new information that significantly advance our understanding of this complex. For these reasons I am quite favorable of publication, though a couple of issues might be considered in revision.

- 1) Does the interaction of BRD9 with the inhibitor lead to loss of any BBAF components? This is important in attributing the targeting to BRD9.
- 2) What are the differences between sites that are affected by the BRD9 inhibitor and those that are refractory? Chromatin modifications, particular TF sites, histone variants? Here, comparisons to known maps would be of interest.

Reviewer #2 (Remarks to the Author):

This manuscript presents evidence for a distinct BAF complex in mouse ESCs that consists of a subset of esBAF subunits as well as bromodomain-containing protein 9 (BRD9) and BRD4-interacting chromatin remodeling associated protein/like (BICRA/L). The authors initially show that BRD9 inhibition leads to the downregulation of numerous genes related to development and mESC biology. A series of biochemical experiments reveal that BRG1, BRD9, and other BAF subunits form a complex (BBAF) that is distinct from esBAF in terms of subunit makeup and genomic localization. ChIPseq and transcriptomic analyses are subsequently used to demonstrate that BBAF modulates KLF4 and SP5 binding and transcriptional activity and thereby positively regulate the naïve pluripotent state. Lastly, it is shown that the BRD9 bromodomain is largely responsible for directing this activity.

The authors present an interesting and convincing study of a distinct BAF complex in ESCs, adding important data regarding the heterogeneity of BAF complexes, potentially a new distinction with respect to naïve vs. primed ESC biology, and chromatin biology in general. Several controls for the loss-of-function data make these results particularly robust, and the picture generated by the authors is likely to spur a good deal of study in each of these sub-disciplines. However, there are key questions that are not addressed, as well as some minor issues; both are addressed below.

Major point 1:

One of the authors' central claims is that the loss/inhibition of BRD9 and thereby BBAF activity negatively regulates the naïve pluripotent state. The presented transcriptional data support this, but more functional data are lacking, and several straightforward experiments could make this position much stronger. For example, naïve and primed mESCs display notably different morphologies in culture. Are these differences observed between BRD9i/shBRD9 cultures and controls? Other differences between the naïve and primed state could also be interrogated, e.g., passaging by single cells or clusters, required culture factors, X-inactivation status, etc. Finally, if the authors believe that BBAF has a role in regulating the naïve phenotype then experiments in that model would be critical in evaluating the BBAF complex. In brief, there are multiple avenues for dramatically strengthening the authors' argument and thereby adding significant functional relevance to the results.

Major point 2:

Although the authors understandably focus on the role of BBAF in the naïve state, one of the most dramatic differences between BBAF and esBAF receives little attention, i.e., the profound dissimilarity in the BRD9xBRG1 and ARID1AxBRG1 ChIP-Seq signals at CTCF sites. Some rather simple bioinformatics analyses would be of great interest to readers and have the potential to augment the impact of the manuscript. For example, i) what percentage of CTCF sites have a higher signal between the two groups? ii) What is special about these differential sites? Are they near TAD boundaries, DEGs, poised/active/super-enhancers, EpiESC genes? The answers to these questions have the potential to both complement and strengthen the described effects.

Minor critiques:

- The motif enrichment data require additional context. For example, what is the percentage of unique BBAF-specific peaks that have the indicated motifs? Three percent? Forty? This is relevant to the interpretation. As regards the methods (and, by extension, the p-values of the results), the regions that were used as the background control in HOMER need to be specified. Given the infinitesimal p-values, I suspect that the background regions were random genomic regions. If so, better/more informative background regions for each analysis may be i) the unique peaks for the other complex, e.g., using the BRG1xARID1A-unique peaks for the background set of the BRG1xBRD9-unique peak analysis; or ii) all BRG1 ChIPseq peaks. At the very least, a set of size-matched open-chromatin regions in mESCs (excluding, of course, the considered peaks themselves) would be a preferable background control to the genome as a whole. In addition, the Methods for this analysis should be fleshed out. For example, the parameters used in the HOMER command should be noted.

- Overall, the manuscript is well written and organized, but a few minor typographical errors should be addressed. For example, the term “acylated” appears several times. The presumption is that “acetylation” is the intended term; if so, these instances should be corrected.

Reviewer #3 (Remarks to the Author):

Summary:

SWI/SNF chromatin remodeler, or BAF complexes, have been previously shown to be essential for the maintenance of stem cells self-renewal and pluripotency. In the manuscript under review, Gatchalian et al. have characterized the role of the bromodomain containing protein BRD9, a known BAF complex subunit, in mESC. The authors have shown mESC to be sensitive to the BRD9 bromodomain (BRD) inhibitor I-BRD9, as well as to knock-down of BRD9 by shRNA. Loss of BRD9 activity resulted in the modulation of a transcriptional regulatory network required for the maintenance of mESC pluripotency. Through biochemical fractionation, the authors then showed that BRD9 colocalized with some but not all of the previously defined esBAF complex subunits. Through IP-WB and IP-MS experiments, they have defined a novel BRD9-containing BAF complex which they termed BBAF. Critically, BBAF contained the BICRAL/BICRA subunits but lacks the esBAF subunits BAF47, ARID1A and BCL11A/B. The authors then shown that BBAF and esBAF localized to distinct chromatin regions through ChIP-seq analysis of BRD9 (BBAF) and ARID1A (esBAF) and furthermore colocalized with distinct pluripotency transcription factors. Loss of BBAF activity achieved by I-BRD9 treatments was found to be different from esBAF alterations caused by KO of ARID1A in that reduced BBAF activity transitioned mESC to a primed state resembling EpiESC. I-BRD9 reduced the association of BRD9 to chromatin as assessed by ChIP-seq and more generally, genes bound by BRD9 also showed a reduced expression upon I-BRD9 treatment. Taken together, the authors claim that BRD9 is part of a mESC specific BAF complex critical to the maintenance of ESC pluripotency.

Critique:

My overall impression of the manuscript by Gatchalian et al. is that it is well written and that the data is in general well-presented and of high quality. The major critique of the work by Gatchalian et al. is that it failed to determine whether BRD9 acts through loss of BRG1 and/or BRD4 activity or in a yet to be define mode-of-action. From previous work by multiple groups, it is known that BRD9 interacts with both BRD4 and BRG1 (e.g. PMID: 21555454). As the loss of BRD4 (PMID: 25263550 and 27498864) or BRG1 activity (PMID: 26702435) results in loss of stem cells pluripotency and ultimately leads to their differentiation (the same phenotypes reported here for BRD9), the authors must place their results within this larger context. Compounding this issue is the fact that I-BRD9 has

some affinity for BRD4 BRDs making experiments using it harder to interpret by themselves. As such, the authors must define whether BRD4 or BRG1 are the root causes for the results they observe using a combination of small molecules inhibitors (JQ1/iBET-151 for BRD4 and PFI-3 for BRG1) and/or genetic approaches. In addition, the authors need to address the points below for a revised manuscript to be acceptable for publication.

Points to addressed in a revised manuscript:

- BRD9 BRD inhibitor. At the moment, at least five distinct specific BRD9 BRD inhibitor are available (I-BRD9, LP99, BI-9564, TP-472 and GNE-375). As I-BRD9 is known to possess some residual affinity for the BRD4 BRDs (Kd ~ 1.3 μ M; PMID: 25856009) and the fact that the authors had to use a high concentration (3.33 μ M) of I-BRD9 in most of their assays (despite the 2nM Kd of I-BRD9 for BRD9 BRD), off-target problems with I-BRD9 is possible. As such, the authors should test multiple BRD9 BRD inhibitors in their survival assay to strengthen their data and show that distinct molecules generate similar trends.
- Is I-BRD9 impacting the BBAF complex or BRD4 interaction network? The impact of I-BRD9 on the protein-protein interactions of BRD9 is not known. Performing IP-MS or IP-WB using antibodies targeting BRD9, BRG1 and BRD4, in the presence or absence of I-BRD9, would allow the authors to determine whether the inhibitor leads to remodeling of protein-protein interactions. Also, in Figure 6, the authors use I-BRD9 to show that ~40% of genes affected by I-BRD9 also show a decreased binding by BRD9 following the treatment. The same analysis should be performed with CHIP-seq data for BRD4 and BRG1. Are the sites resistant to I-BRD9 simply not bound by BRG1 and/or BRD4?
- Are the BRD9-responsive genes identified in Figure 1 also BRD4-dependent genes?
- Is BBAF the same complex as the recently identified GBAF complex? Recently, Alpsy and Dykhuizen reported the identification of the GBAF complex, containing BRG1, BRD9, BRICA, BICRAL, BAF155, BAF60, BAF53a, SS18 and ACTB, from numerous cancer cell lines (PMID: 29374058). As GBAF is extremely similar to the BBAF reported here, the authors should determine whether BRD9 purification from non-mESC allows them to identify the complex as well. To me, it appears that the GBAF and BBAF complexes are the same with their differences due to the different bait used for their purifications.
- Proteomics data. The complete proteomics results should be attached to the manuscript under review. In addition, the raw unprocessed data (.raw files in this case) should be deposited in one of the numerous free repository currently available (<http://www.proteomexchange.org/>; <https://massive.ucsd.edu/ProteoSAFe/static/massive.jsp>; etc.) to allow their reanalysis by others but

also to safeguard them. As the moment, it is not possible for this reviewer to assess the quality of the IP-MS data presented in the manuscript since Figure 2C is only showing selected proteins rather than the complete data. Also, rather than showing log₂ FC, the authors should employ one of the various algorithm available (compass, SAINT, MaxQuant, etc.) to determine which proteins are statistically enriched in their BRD9 IP compared to the IgG controls. In particular, it is essential to be able to determine whether BRD4 is associated with BRD9 in mESC. Finally, the authors needs to state whether replicates was performed for these experiments.

Minor points:

- The blots in supplemental figure 1 should be quantified to help reader determined which shRNA is the most effective at knocking down BRD9.
- If fractions used in Figure 2 are still available, the figure would benefit of blotting for BRD4, BRM (BRG1 homologue), BICRA and BICRAL.
- In figure 2B and 2D, IgG IP control are needed to ensure that co-purified proteins identified are specific, especially subunits that are found in all lanes (e.g. BAF155, BAF60a, etc.).
- To minimize the signal generated by IgG bands in Figure 2B for BAF57, the authors should look into using secondary antibodies that only recognize folded antibodies such as the TrueBlot reagents.
- What is the overlap between the binding sites of BRD9, ARID1A and BRG1 with BRD4 in Figure 3?
- In Figure 6, a simple fractionation experiment allowing fractions of the cytoplasm, nucleoplasm and chromatin to be collected would enable the author to determine whether BRD9 is displaced away from chromatin upon I-BRD9.
- The source of I-BRD9 and the product # for the ES-qualified serum should be stated.

We thank the reviewers for their insightful comments and suggestions. We hope that the revised manuscript and the following responses shed more light into the function of BRD9-BAF complex in stem cell pluripotency. We have provided additional experiments using other inhibitors and alternative methods (chromatin fractionation) to strengthen our initial claims. In addition, we have added experiments and a figure incorporating new functional data in serum/LIF (Figure 5) and 2i conditions (Figure 8). We have also added an additional figure elucidating the mechanism of cooperation/targeting of BRD9-BAF complexes by BRD4 (Figure 7). Finally, we have added two authors, Jesse Dixon and Dong-Sung Lee, who performed the analysis of BRD9/ARID1A/BRG1 binding at TAD boundaries. Please note that we now refer to the complex as the GBAF complex and BICRA/L as GLTSCR1/L. Below is a point-by-point response:

Reviewer #1

Overall, I find this a really interesting paper. The novelty of the first section is diminished by the recent JBC paper, but the middle and end of the paper involving the genomics analyses are novel and interesting, and provide several pieces of new information that significantly advance our understanding of this complex. For these reasons I am quite favorable of publication, though a couple of issues might be considered in revision.

We thank the reviewer for these comments and are glad that he/she finds our manuscript interesting and is favorable of publication. In response to the first point, we performed additional biochemical experiments in ESCs (Figure 2A-C) and HCT116 cells (Supplementary Figure 2) to determine if the complex we had identified is indeed identical to the GBAF complex described by Alpsy and Dykhuizen [1]. Given very similar findings, we now refer to the BBAF complex as the GBAF complex for consistency. It should be noted that we are the first to perform IP-mass spectrometry of this novel complex to define its components in entirety. In addition, as pointed out by the reviewer, we go well beyond biochemical characterization to define the genomic and functional properties of the GBAF complex and provide evidence for its specific role in regulating naïve pluripotency in ESCs. Finally, we are the first to show a role for targeting for any chromatin binding domain within the BAF complex.

1) Does the interaction of BRD9 with the inhibitor lead to loss of any BBAF components? This is important in attributing the targeting to BRD9.

This is an important point raised by Reviewer 1. We addressed this by performing western blot on chromatin-bound BRG1 and ChIP-seq for BRG1 with or without I-BRD9 inhibition. We did not see a difference in global levels of chromatin-bound BRG1 by western blot between DMSO and I-BRD9 treated cells (data not shown). We estimated from quantitation of the glycerol gradient fractions that only ~17% of BRG1 is in GBAF complexes, so at most, we would expect to see a 17% decrease in BRG1 levels from chromatin, which may not be detectable at the global level. We did however see an overall decrease in BRG1 binding at sites annotated to I-BRD9 DEGs (Figure 6I, 6J). BRG1 binding was significantly decreased (FC>1.5, Poisson p value<0.0001) at some, but not all of the sites where BRD9 binding was lost with I-BRD9 treatment. This may be due to preexisting esBAF or PBAF binding at these sites, which is not affected by I-

BRD9, or possibly due to compensatory binding by BAF/PBAF complexes in the absence of BRD9, as has been reported previously in the context of BAF subunit mutations [2]. To address whether I-BRD9 leads to loss of GBAF-specific components, we attempted to perform ChIP against the GBAF-unique subunit GLTSCR1 in ESCs with or without I-BRD9 followed by quantitative PCR. We observed a decrease in GLTSCR1 binding at the *Nanog* enhancer with I-BRD9 treatment (Rebuttal Figure 1). However, the results were not robust upon replication due to insufficient antibody quality. We did not attempt to perform ChIP-qPCR for GLTSCR1L as the antibody against GLTSCR1L does not work for IP. In any case, the reduced binding of BRG1 in I-BRD9 treated cells is consistent with a role for the BRD9 BD in targeting GBAF complexes.

2) What are the differences between sites that are affected by the BRD9 inhibitor and those that are refractory? Chromatin modifications, particular TF sites, histone variants? Here, comparisons to known maps would be of interest.

This is an important point raised by both Reviewer 1 and Reviewer 3. We performed a comprehensive analysis of publicly available datasets and datasets generated in house for differential enrichment between I-BRD9-sensitive and refractory sites (Rebuttal Figure 2). The two classes are similarly enriched for several common histone modifications. There is higher enrichment for ARID1A and the pluripotency TFs OCT4, SOX2, and NANOG at sites that are refractory to I-BRD9, which suggests that there is higher retention of GBAF when esBAF is co-bound. KLF4 was marginally enriched at I-BRD9 sensitive sites. However, we did not observe higher enrichment for CTCF or Sp5 at I-BRD9-sensitive sites. In addition, BRD4 was similarly enriched between the two classes. Our data suggest that BRD9 is targeted via recognition of an acetylated residue on BRD4, and thus we predict that an acetylated form of BRD4 would be enriched at I-BRD9-sensitive sites. We cannot however rule out the possibility that there is a specific histone mark that is recognized by the BRD9 BD in some contexts.

Reviewer 2:

The authors present an interesting and convincing study of a distinct BAF complex in ESCs, adding important data regarding the heterogeneity of BAF complexes, potentially a new distinction with respect to naïve vs. primed ESC biology, and chromatin biology in general. Several controls for the loss-of-function data make these results particularly robust, and the picture generated by the authors is likely to spur a good deal of study in each of these sub-disciplines. However, there are key questions that are not addressed, as well as some minor issues; both are addressed below.

We thank the reviewer for appreciating that our work represents a significant advance in the fields of chromatin biology and BAF complex biology. We are glad the reviewer finds the work thorough, well-controlled, and convincing. We agree that our work is an important study in understanding how heterogeneity in BAF complexes allows for greater control over complex regulatory networks of pluripotency.

Major point 1:

One of the authors' central claims is that the loss/inhibition of BRD9 and thereby BBAF activity negatively regulates the naïve pluripotent state. The presented transcriptional data support this, but more functional data are lacking, and several straightforward experiments could make this position much stronger. For example, naïve and primed mESCs display notably different morphologies in culture. Are these differences

observed between BRD9i/shBRD9 cultures and controls? Other differences between the naïve and primed state could also be interrogated, e.g., passaging by single cells or clusters, required culture factors, X-inactivation status, etc. Finally, if the authors believe that BBAF has a role in regulating the naïve phenotype then experiments in that model would be critical in evaluating the BBAF complex. In brief, there are multiple avenues for dramatically strengthening the authors' argument and thereby adding significant functional relevance to the results.

This is an excellent point. We have now included several experiments to address these concerns, including clonogenic assays, alkaline phosphatase quantitation, and morphological characterization. We confirmed that treatment with I-BRD9 or BI-9564 gave rise to cells with several characteristics of the primed state, including less clonogenic potential after single-cell passage (Figure 5C), fewer AP+ clones (Figure 5D), and flatter morphology (Figure 5B) in a concentration-dependent manner [3-5]. Furthermore, we have added Figure 8, where we address the role of BRD9 in the 2i model of naïve pluripotency. Specifically, we found that I-BRD9 had no effect on self-renewal (Figure 8A) or expression of pluripotency regulators (Figure 8B-D) in 2i culture conditions, which force cells to adopt the naïve or ground state by inhibiting pro-differentiation signaling. These data are consistent with a role for BRD9 in promoting naïve pluripotency in conjunction with BRD4, as BRD4 was recently found to be dispensable for naïve pluripotency in 2i conditions [6]. These new data address the reviewer's request for functional data and validate our previous claims that BRD9 is required to maintain the naïve pluripotent state in serum/LIF conditions.

Major point 2:

Although the authors understandably focus on the role of BBAF in the naïve state, one of the most dramatic differences between BBAF and esBAF receives little attention, i.e., the profound dissimilarity in the BRD9xBRG1 and ARID1AxBRG1 ChIP-Seq signals at CTCF sites. Some rather simple bioinformatics analyses would be of great interest to readers and have the potential to augment the impact of the manuscript. For example, i) what percentage of CTCF sites have a higher signal between the two groups? ii) What is special about these differential sites? Are they near TAD boundaries, DEGs, poised/active/super-enhancers, EpiESC genes? The answers to these questions have the potential to both complement and strengthen the described effects.

We too find this difference in binding very intriguing. In response to i), we observed greater overlap of CTCF with GBAF than esBAF, with 18% and 0.5% of CTCF sites being bound by GBAF and esBAF, respectively. As suggested by the reviewer in ii), we teamed up with expert Jesse Dixon (Dixon Nature 2012) and his postdoctoral associate Dong-Sung Lee and found that BRD9 localizes much more strongly than ARID1A to TAD boundaries (Figure 3E), where CTCF is also strongly enriched. We further examined the enrichment of histone modifications and enhancer classes at CTCF sites

that are either bound by esBAF or GBAF (Rebuttal Figure 3A). Although there are only about 300 esBAF-bound CTCF sites, these regions are enriched for H3K4me, H3K4me3, and H3K27ac, as well as active and super enhancers. In contrast, we observed an enrichment of H3K4me3 and H3K27ac at GBAF-bound CTCF sites, consistent with an enrichment of BRD9 at promoters and TAD boundaries (Dixon Nature 2012). Furthermore, 60% of I-BRD9 DEGs and 53% EpiESC genes are bound by overlapping CTCF and BRD9, pointing to a cooperative role for CTCF in BRD9's function in regulating gene expression (Rebuttal Figure 3B). Further studies are geared toward understanding the role of CTCF in regulating BRD9-dependent genes as well as the potential collaboration between GBAF and CTCF in regulating chromatin domain organization. We appreciate the reviewer's understanding concerning our choice to focus on the cooperation between BRD4 and BRD9 in regulating naïve pluripotency in the current manuscript.

Minor critiques:

- *The motif enrichment data require additional context. For example, what is the percentage of unique BBAF-specific peaks that have the indicated motifs? Three percent? Forty? This is relevant to the interpretation. As regards the methods (and, by extension, the p-values of the results), the regions that were used as the background control in HOMER need to be specified. Given the infinitesimal p-values, I suspect that the background regions were random genomic regions. If so, better/more informative background regions for each analysis may be i) the unique peaks for the other complex,*

e.g., using the BRG1xARID1A-unique peaks for the background set of the BRG1xBRD9-unique peak analysis; or ii) all BRG1 ChIPseq peaks. At the very least, a set of size-matched open-chromatin regions in mESCs (excluding, of course, the considered peaks themselves) would be a preferable background control to the genome as a whole. In addition, the Methods for this analysis should be fleshed out. For example, the parameters used in the HOMER command should be noted.

As suggested by the reviewer, we have specified in the Methods section that we used GC content-matched genomic regions as background for the motif analysis and added the fragment size and motif length parameters used for the command. We have also included the % of target and background sequences that contain each indicated motif for GBAF- and esBAF-specific sites in Figure 4A. When we performed the motif analysis against all BRG1 peaks as background, the resulting motifs are consistent with what we observed using random genomic regions as background. We found that ARID1AxBRG1-unique peaks are still enriched for the pluripotency TF motif (OCT4-SOX2-TCF-NANOG) and motifs of different SOX family members (Rebuttal Figure 4A). For BRD9xBRG1-unique peaks, CTCF and CTCFL motifs are still the most enriched, with the Sp1 motif ranking as # 8 on the list (Rebuttal Figure 4B). RONIN, GFY, GFY-Staf3 are found in less than 5% of target sequences but are rare in the background. Notably, RONIN has also been implicated in maintenance of pluripotency in serum/LIF conditions [7, 8]. The GFY motifs are not as informative because the factors bound to these sequences are either unknown or there is not enough data to support that such factors bind to the motifs (<http://homer.ucsd.edu/homer/ngs/peakMotifs.html>). We favor using the results of the first motif analysis against random genomic regions because of the strong correlation between BRD9-dependent genes and EpiESC genes, which are largely regulated by the naïve pluripotency TFs Sp5 and KLF4.

- Overall, the manuscript is well written and organized, but a few minor typographical errors should be addressed. For example, the term “acylated” appears several times. The presumption is that “acetylation” is the intended term; if so, these instances should be corrected.

Acylation refers to the addition of an acyl group and includes modifications such as acetylation, but also butyrylation, crotonylation and others. BRD9 has been reported to bind acetyl, propionyl, butyryl modifications in vitro [9]. The specificity of BRD9 in vivo is not known. To be inclusive, we used the word ‘acylation’ in the first draft. For simplicity and to avoid confusion, we have used ‘acetylation’ in the revision.

Reviewer #3

Critique:

My overall impression of the manuscript by Gatchalian et al. is that it is well written and that the data is in general well-presented and of high quality. The major critique of the work by Gatchalian et al. is that it failed to determine whether BRD9 acts through loss of BRG1 and/or BRD4 activity or in a yet to be define mode-of-action. From previous work

by multiple groups, it is known that BRD9 interacts with both BRD4 and BRG1 (e.g. PMID: 21555454). As the loss of BRD4 (PMID: 25263550 and 27498864) or BRG1 activity (PMID: 26702435) results in loss of stem cells pluripotency and ultimately leads to their differentiation (the same phenotypes reported here for BRD9), the authors must place their results within this larger context. Compounding this issue is the fact that I-BRD9 has some affinity for BRD4 BRDs making experiments using it harder to interpret by themselves. As such, the authors must define whether BRD4 or BRG1 are the root causes for the results they observe using a combination of small molecules inhibitors (JQ1/iBET-151 for BRD4 and PFI-3 for BRG1) and/or genetic approaches. In addition, the authors need to address the points below for a revised manuscript to be acceptable for publication.

We thank the reviewer for their critique. We have now significantly extended our findings to demonstrate that BRD9 acts through the described GBAF complex in cooperation with BRD4. We find that the effects of treating with I-BRD9 are similar to the effects of inhibiting or deleting BRG1 or BRD4 with regard to pluripotency and transcriptional changes. This is not due to off-target effects of I-BRD9, but rather through the recruitment of GBAF complexes by BRD4. Below we describe the specific experiments we have done to address the reviewer's concerns and place our results within the larger framework of epigenetic regulation of pluripotency.

Points to addressed in a revised manuscript:

- BRD9 BRD inhibitor. At the moment, at least five distinct specific BRD9 BRD inhibitor are available (I-BRD9, LP99, BI-9564, TP-472 and GNE-375). As I-BRD9 is known to possess some residual affinity for the BRD4 BRDs (Kd ~ 1.3 μ M; PMID: 25856009) and the fact that the authors had to use a high concentration (3.33 μ M) of I-BRD9 in most of their assays (despite the 2nM Kd of I-BRD9 for BRD9 BRD), off-target problems with I-BRD9 is possible. As such, the authors should test multiple BRD9 BRD inhibitors in their survival assay to strengthen their data and show that distinct molecules generate similar trends.

We have now used several BRD9 inhibitors, including I-BRD9, TP472, and BI-9564 in survival assays (Figure 1A, Supplementary Figure 1A) and functional assays (Figure 5C, D). We found that BI-9564 in particular robustly affected survival and clonogenic potential even at lower concentrations. Given that we saw similar results with all BRD9 inhibitors tested, we are confident that the effects of I-BRD9 are specific to BRD9 inhibition. We have also included survival data with the BRG1 inhibitor PFI-3 (Supplementary Figure 1C).

- Is I-BRD9 impacting the BBAF complex or BRD4 interaction network? The impact of I-BRD9 on the protein-protein interactions of BRD9 is not known. Performing IP-MS or IP-WB using antibodies targeting BRD9, BRG1 and BRD4, in the presence or absence of I-BRD9, would allow the authors to determine whether the inhibitor leads to remodeling of protein-protein interactions.

These are great suggestions. We have now included an additional figure to show that I-BRD9 is impacting the BRD4 interaction network. Specifically, we have performed IP-WB of BRD9 under different conditions to show that BRD9 interacts with BRD4 (Supplementary Figure 4C). This interaction is strengthened by pre-treating the cells with TSA, and diminished in the presence of I-BRD9 (Figure 7C). In contrast, I-BRD9 has no effect on the interaction of GBAF components (Supplementary Figure 3). We conclude that the interactions between BRD9 and other GBAF components are not dependent on the BRD9 BD. In contrast, the interaction between BRD9 and BRD4 is dependent on the BRD9 BD, likely through the recognition of a post-translational modification on BRD4. This is consistent with our biochemical data indicating that the interaction between BRD9 and BRD4 can be disrupted in high stringency wash conditions (Supplementary Figure 4C), and strengthened with TSA treatment (Figure 7C). Furthermore, we find that BRD9, BRG1, and BRD4 are co-localized at nearly 19,000 sites by ChIP-seq (Figure 7B). Finally, consistent with cooperative recruitment of BRD9 and BRD4, I-BRD9 and JQ1 result in highly correlated changes in transcription (Figure 7A).

Also, in Figure 6, the authors use I-BRD9 to show that ~40% of genes affected by I-BRD9 also show a decreased binding by BRD9 following the treatment. The same analysis should be performed with ChIP-seq data for BRD4 and BRG1.

We have now performed ChIP-seq with BRG1 and BRD4 in I-BRD9-treated cells. We find that 50% of I-BRD9 DEGs lose BRD9 with I-BRD9 treatment (Figure 7G). In addition, we find that BRG1 binding is reduced at I-BRD9 DEGs, albeit less so (Figure 6I,J). In contrast, BRD4 was not displaced from chromatin following I-BRD9 treatment (Supplementary Figure 4A). Very few BRD4 sites were lost in BRD4 ChIP-seq of I-BRD9 treated cells, and as many were gained (Figure 7D). Finally, there was essentially no change in BRD4 binding at I-BRD9 DEGs (Rebuttal Figure 5) and <3% of the I-BRD9 DEGs lost BRD4 binding following I-BRD9 treatment. This demonstrates that I-BRD9 does not act non-specifically to displace BRD4 from chromatin. Rather, we find that BRD9 is displaced by treating with the BRD4 inhibitor, JQ1, at nearly 13,000 sites (Figure 7E). In fact, BRD9 binding appears to be sensitive to both I-BRD9 and JQ1 at nearly 7,000 sites including those at I-BRD9 DEGs (Figure 7F,G). Because JQ1 has no affinity for BRD9 [10], these data suggest that BRD4 acts to target BRD9 via an interaction that is dependent on the BRD9 BD and likely auxiliary interactions with GLTSCR1, as previously reported [11].

Are the sites resistant to I-BRD9 simply not bound by BRG1 and/or BRD4?

Please see response to Reviewer 1, Rebuttal Figure 2. We have performed enrichment analyses against BRD9 sites that are sensitive or refractory to I-BRD9 treatment for histone modifications and other factors. We do not find that either BRG1 or BRD4 are specifically enriched in either class. We would not expect BRG1 to be differentially bound as our biochemical data indicates that BRD9 can be depleted from nuclear lysates with an antibody against BRG1 (Figure 2E) and is associated with BRG1 in up to 2.5M urea (Figure 2F). This demonstrates that BRD9 is a bona fide BAF subunit and does not exist in a free form, unassociated with GBAF. Therefore, all GBAF sites, whether they are sensitive to I-BRD9 or not, should have BRG1. Indeed, we find very strong overlap between BRD9 and BRG1 binding sites (Figure 3A). Since BRD4 plays a role in targeting BRD9, we might have expected to find an enrichment for BRD4 in the I-BRD9 sensitive sites. However, BRD9 and BRD4 are not exclusively co-localized (Figure 7B). This suggests that there may be other factors, or more likely a specific acetylated form of BRD4 that specifies I-BRD9-dependent targeting of BRD9. In addition, BRD9 may interact directly with post-translational modifications on histone proteins in an I-BRD9 dependent fashion. Indeed, our data suggest that BRD9 can be targeted via BRD4 BD-dependent and independent mechanisms (Figure 7G).

- Are the BRD9-responsive genes identified in Figure 1 also BRD4-dependent genes?

Yes, they are. Please see Figure 7A, as well as Supplementary Figure 4B. As outlined above, our data demonstrates that the correlation in I-BRD9 and JQ1 conditions is due to the loss of BRD9 in both treatments and the cooperative action of GBAF and BRD4 in regulating naïve pluripotency genes (Figure 7E-G).

- Is BBAF the same complex as the recently identified GBAF complex? Recently, Alpsy and Dykhuizen reported the identification of the GBAF complex, containing BRG1, BRD9, BRICA, BICRAL, BAF155, BAF60, BAF53a, SS18 and ACTB, from

numerous cancer cell lines (PMID: 29374058). As GBAF is extremely similar to the BBAF reported here, the authors should determine whether BRD9 purification from non-mESC allows them to identify the complex as well. To me, it appears that the GBAF and BBAF complexes are the same with their differences due to the different bait used for their purifications.

We have now performed additional experiments in HCT116 cells (Supplementary Figure 2) and find a small GBAF complex with identical properties to the ESC GBAF complex. The complex we identified does appear to have similar properties to the one described by Alpsy and Dykhuizen and thus, we refer to the complex as the GBAF complex in the revised manuscript. Importantly, we have performed IP-MS of this complex (they did not) to define its components in full.

- Proteomics data. The complete proteomics results should be attached to the manuscript under review. In addition, the raw unprocessed data (.raw files in this case) should be deposited in one of the numerous free repository currently available (<http://www.proteomexchange.org/>; <https://massive.ucsd.edu/ProteoSAFe/static/massive.jsp>; etc.) to allow their reanalysis by others but also to safeguard them. As the moment, it is not possible for this reviewer to assess the quality of the IP-MS data presented in the manuscript since Figure 2C is only showing selected proteins rather than the complete data. Also, rather than showing log2 FC, the authors should employ one of the various algorithm available (compass, SAINT, MaxQuant, etc.) to determine which proteins are statistically enriched in their BRD9 IP compared to the IgG controls. In particular, it is essential to be able to determine whether BRD4 is associated with BRD9 in mESC. Finally, the authors needs to state whether replicates was performed for these experiments.

Thank you for these suggestions. We performed two biological replicates for the IP-MS experiment. However, the first experiment's IgG IP contained many non-specific binding proteins. We subsequently switched our protocol from using cross-linked agarose beads to Dynabeads (Invitrogen) and our second experiment was appreciably cleaner. Since the two experiments were not processed identically, we did not perform statistical analyses. Two technical replicates were ran for the second experiment and we used PatternLab [12] to determine which proteins were significantly enriched in the BRD9 IP compared to the IgG IP. The results are plotted in Figure 2A. We did not recover BRD4 in the IP-MS because the IP was done under stringent wash conditions, which we show can disrupt the BRD9:BRD4 interaction (Supplementary Figure 4C). We have included a list from the proteomics results for reference (Supplementary Table 1). In addition, the raw files have been deposited into the ProteomeXchange archive with the following ID PXD010670. For reviewer access, please use the following credentials. username: reviewer77293@ebi.ac.uk password: IVVbcsQy

Minor points:

- The blots in supplemental figure 1 should be quantified to help reader determined which shRNA is the most effective at knocking down BRD9.

The knockdown efficiency with sh*Brd9* has been quantified and included in Supplementary Figure 1B.

- If fractions used in Figure 2 are still available, the figure would benefit of blotting for BRD4, BRM (BRG1 homologue), BICRA and BICRAL.

We have blotted for BRD4 and GLTSCR1 (BICRA) and included them in Figure 2B. We tried blotting for GLTSCR1L (BICRAL) but because the antibody detection isn't superb and the gradient fractions are quite dilute, we could not detect it. BRM is known to be expressed at low levels in mESCs [13] and we believe BRG1 to be the dominant ATPase subunit in this cell type. However, we did detect BRM in our IP-MS (Supplementary Table 1), suggesting that BRD9 can associate with either ATPase.

- In figure 2B and 2D, IgG IP control are needed to ensure that co-purified proteins identified are specific, especially subunits that are found in all lanes (e.g. BAF155, BAF60a, etc.).

An IgG IP control has now been included in Figure 2C. Thank you for catching this.

- To minimize the signal generated by IgG bands in Figure 2B for BAF57, the authors should look into using secondary antibodies that only recognize folded antibodies such as the TrueBlot reagents.

Thank you for this excellent suggestion. We performed western blot detection for BAF57 and BAF53A with an Abcam Veriblot secondary antibody (#ab131366) and the results are included in Figure 2C. The final image incorporates different exposures on film, which were done to ensure detection for the different IP conditions. We included the original scans for the reviewers and readers' inspection (Supplementary Material).

- What is the overlap between the binding sites of BRD9, ARID1A and BRG1 with BRD4 in Figure 3?

We have included the Venn diagrams for BRG1, BRD9 and BRD4 in Figure 7B and for BRG1, ARID1A and BRD4 in Supplementary Figure 4D. We find that BRD4 co-localizes with BRD9 and BRG1 at 43% of BRD4 binding sites, but only 25% of BRD4 binding sites are bound by ARID1A. Additionally, we observed an interaction between BRD9 and BRD4 (Figure 7C, Supplementary Figure 4C).

- In Figure 6, a simple fractionation experiment allowing fractions of the cytoplasm, nucleoplasm and chromatin to be collected would enable the author to determine whether BRD9 is displaced away from chromatin upon I-BRD9.

We have added the results of the fractionation assay with or without I-BRD9 in Figure 6A. BRD9 is indeed displaced from chromatin with I-BRD9, while BRD4 is not (Supplementary Figure 4A), demonstrating the on-target effects of I-BRD9.

- *The source of I-BRD9 and the product # for the ES-qualified serum should be stated.*

Agreed. These have now been incorporated into the Materials and Methods.

References

1. Alpsy, A. and E.C. Dykhuizen, *Glioma tumor suppressor candidate region gene 1 (GLTSCR1) and its paralog GLTSCR1-like form SWI/SNF chromatin remodeling subcomplexes*. J Biol Chem, 2018. **293**(11): p. 3892-3903.
2. Chandler, R.L., et al., *ARID1a-DNA interactions are required for promoter occupancy by SWI/SNF*. Mol Cell Biol, 2013. **33**(2): p. 265-80.
3. Tesar, P.J., et al., *New cell lines from mouse epiblast share defining features with human embryonic stem cells*. Nature, 2007. **448**(7150): p. 196-9.
4. Brons, I.G., et al., *Derivation of pluripotent epiblast stem cells from mammalian embryos*. Nature, 2007. **448**(7150): p. 191-5.
5. Nichols, J. and A. Smith, *Naive and primed pluripotent states*. Cell Stem Cell, 2009. **4**(6): p. 487-92.
6. Finley, L.W.S., et al., *Pluripotency transcription factors and Tet1/2 maintain Brd4-independent stem cell identity*. Nat Cell Biol, 2018. **20**(5): p. 565-574.
7. Dejosez, M., et al., *Ronin is essential for embryogenesis and the pluripotency of mouse embryonic stem cells*. Cell, 2008. **133**(7): p. 1162-74.
8. Dejosez, M., et al., *Ronin/Hcf-1 binds to a hyperconserved enhancer element and regulates genes involved in the growth of embryonic stem cells*. Genes Dev, 2010. **24**(14): p. 1479-84.
9. Flynn, E.M., et al., *A Subset of Human Bromodomains Recognizes Butyryllysine and Crotonyllysine Histone Peptide Modifications*. Structure, 2015. **23**(10): p. 1801-1814.
10. Filippakopoulos, P., et al., *Selective inhibition of BET bromodomains*. Nature, 2010. **468**(7327): p. 1067-73.
11. Rahman, S., et al., *The Brd4 extraterminal domain confers transcription activation independent of pTEFb by recruiting multiple proteins, including NSD3*. Mol Cell Biol, 2011. **31**(13): p. 2641-52.
12. Carvalho, P.C., et al., *Integrated analysis of shotgun proteomic data with PatternLab for proteomics 4.0*. Nat Protoc, 2016. **11**(1): p. 102-17.
13. Ho, L., et al., *An embryonic stem cell chromatin remodeling complex, esBAF, is essential for embryonic stem cell self-renewal and pluripotency*. Proc Natl Acad Sci U S A, 2009. **106**(13): p. 5181-6.

Reviewer #1 (Remarks to the Author):

Review of revised NCOMMS-18-05631A by Gatchalian et al., (D. Hargreaves lab).

In their revision, the authors addressed in some detail each of the main requests I made in my original review. I was especially interested in the expanded work on the impact of the BRD9 inhibitor and its relationship to BRD4. Overall, the revisions that I and the other reviewers requested helped to improve what was already a good manuscript. This work will be of interest to many in the chromatin remodeling and bromodomain/BRD fields - and I support publication.

Reviewer #3 (Remarks to the Author):

In their revised manuscript and rebuttal letter, Gatchalian et al. have been able to address most the concerns initially raised by all reviewers. By placing their results in the larger context of other BRD-containing proteins, they have definitely improved their manuscript. For this, the authors should be commended.

As of now, my only concern reside in the Western blots shown in Figure 6A and 7C. In both cases, the authors make big claims on subtle changes in band intensity (especially so for Figure 7C). Personally, I would suggest to the authors to alter some of the text describing these sections to better reflect this situation.

Despite this small concern, I am in support of publishing the revised manuscript by Gatchalian et al. and wish to congratulate the authors for their great work.

REVIEWERS' COMMENTS:

Reviewer #1 (Remarks to the Author):

Review of revised NCOMMS-18-05631A by Gatchalian et al., (D. Hargreaves lab).

In their revision, the authors addressed in some detail each of the main requests I made in my original review. I was especially interested in the expanded work on the impact of the BRD9 inhibitor and its relationship to BRD4. Overall, the revisions that I and the other reviewers requested helped to improve what was already a good manuscript. This work will be of interest to many in the chromatin remodeling and bromodomain/BRD fields - and I support publication.

Reviewer #3 (Remarks to the Author):

In their revised manuscript and rebuttal letter, Gatchalian et al. have been able to address most the concerns initially raised by all reviewers. By placing their results in the larger context of other BRD-containing proteins, they have definitely improved their manuscript. For this, the authors should be commended.

As of now, my only concern reside in the Western blots shown in Figure 6A and 7C. In both cases, the authors make big claims on subtle changes in band intensity (especially so for Figure 7C). Personally, I would suggest to the authors to alter some of the text describing these sections to better reflect this situation.

Despite this small concern, I am in support of publishing the revised manuscript by Gatchalian et al. and wish to congratulate the authors for their great work.

We thank the reviewers both for their time and for these complimentary remarks. We agree that the manuscript was greatly improved by their suggestions. In response to concerns raised by Reviewer #3, we have altered the text describing Figures 6a and 7c to reflect the changes observed in the data. We have also included an additional experiment for the Western blot quantitation in Figure 7d, where we observed a 3 fold increase in BRD4 binding to BRD9 in TSA conditions, and a 12 fold reduction in this binding with addition of I-BRD9.